# The coenzyme A precursor pantethine enhances antitumor immunity in sarcoma

Richard Miallot[1], Virginie Millet[1], Anais Roger[1], Romain Fenouil[1], Catherine Tardivel[2], Jean-Charles Martin[2], Laetitia Shintu[3], Paul Berchard[4], Juliane Sousa Lanza[1], Bernard Malissen[1,5], Sandrine Henri[1], Sophie Ugolini[1], Aurélie Dutour[4], Pascal Finetti[6], François Bertucci[6,7], Jean-Yves Blay[4,8], Franck Galland[1], Philippe Naquet[1]

**The tumor microenvironment is a dynamic network of stromal, cancer, and immune cells that interact and compete for resources. We have previously identified the Vanin1 pathway as a tumor suppressor of sarcoma development via vitamin B5 and coenzyme A regeneration. Using an aggressive sarcoma cell line that lacks Vnn1 expression, we showed that the administration of pantethine, a vitamin B5 precursor, attenuates tumor growth in immunocompetent but not nude mice. Pantethine boosts antitumor immunity, including the polarization of myeloid and dendritic cells towards enhanced IFNγ-driven antigen presentation pathways and improved the development of hypermetabolic effector CD8+ T cells endowed with potential antitumor activity. At later stages of treatment, the effect of pantethine was limited by the development of immune cell exhaustion. Nevertheless, its activity was comparable with that of anti-PD1 treatment in sensitive tumors. In humans, *VNN1* expression correlates with improved survival and immune cell infiltration in soft-tissue sarcomas, but not in osteosarcomas. Pantethine could be a potential therapeutic immunoadjuvant for the development of antitumor immunity.**

## Introduction

Soft-tissue sarcomas (STSs) are heterogeneous subtypes of tumors with distinct origins and genomic profiles (Fletcher, 2014). To stratify them and define prognostic markers, large-scale approaches have revealed how histio- and genotypes are associated with distinct epigenetic, metabolic or immune traits (Cancer Genome Atlas Research Network et al, 2017). A bioinformatics analysis of TCGA database centered on metabolic signatures showed that, whereas enhanced glycolysis was a hallmark of sarcomas, the degree of oxidative phosphorylation varied significantly amongst histiotypes

(Miallot et al, 2021). Mitochondrial activity conditions metabolic plasticity (Masoud et al, 2020), adaptation to stress, and progression towards aggressive phenotypes (Senyilmaz & Teleman, 2015). Along this line, Vanin1 pantetheinase that regulates the generation of the coenzyme A (CoA) precursor pantothenate (Pan), also known as vitamin B5 (VitB5) (Naquet et al, 2020; Millet et al, 2022), acts as a negative regulator of spontaneous sarcoma development in a p16/p19-deficient tumor-prone mouse model (Giessner et al, 2018). Accordingly, *VNN1* expression tends to be lost in advanced sarcomas and a in a limited set of human sarcomas, *VNN1* transcript levels are associated with improved metastasis-free survival (MFS). Furthermore, in mouse models, Vnn1-expressing tumor lines are less aggressive, show improved mitochondrial activity, and are enriched in immune transcriptional signatures (Giessner et al, 2018). So, we reasoned that by providing pharmacologically the metabolites released by Vnn1 activity, one might recapitulate their antitumor properties through modulation of CoA-dependent mitochondrial activity and redox metabolism. Ultimately, this therapy might impact tumor progression and/or immune cell metabolism and functionality. The development of immune responses is influenced by metabolic cues released within the tumor microenvironment (Naquet et al, 2016; de la Cruz-López et al, 2019). In a stressful context, chronically activated immune cells compete for nutrients with tumor cells (Chang et al, 2015; Geltink et al, 2018; Reinfeld et al, 2021). Furthermore, hypoxic conditions can enhance the development of effector T cell responses (Doedens et al, 2013; Gropper et al, 2017) but also drive CD8+ T cells towards exhaustion (Scharping et al, 2021). Exhausted CD8+ lymphocytes display abnormal mitochondrial biogenesis (Scharping et al, 2016), with a reduction in cristae structure and mitochondria/endoplasmic reticulum (ER) contacts (Yu et al, 2020). Accordingly, PGC1α-transfection in CD8+ T cells enhanced the development of antitumor CD8+ tumor-infiltrating lymphocytes (TILs) (LeBleu et al, 2014). As the loss of mitochondrial fitness contributes to exhaustion

[1]INSERM, CNRS, Centre D'Immunologie de Marseille-Luminy, Aix-Marseille Université, Marseille, France [2]INRAE, INSERM, C2VN, Aix Marseille Université, Marseille, France [3]CNRS, Centrale Marseille, ISM2, Aix Marseille Université, Marseille, France [4]INSERM 1052, CNRS 5286, Cancer Research Center of Lyon (CRCL), Childhood Cancers and Cell Death Laboratory, Lyon, France [5]INSERM, CNRS, Centre D'Immunophénomique (CIPHE), Aix Marseille Université, Marseille, France [6]INSERM, CNRS, Centre de Recherche en Cancérologie de Marseille (CRCM), Institut Paoli-Calmettes (IPC), Laboratory of Predictive Oncology, Aix-Marseille Université, Marseille, France [7]Institut Paoli-Calmettes, Department of Medical Oncology, Marseille, France [8]UNICANCER Centre Léon Bérard, Department of Medicine, Université Lyon I, Lyon, France

Correspondence: naquet@ciml.univ-mrs.fr; richardmiallot@gmail.com

(Soto-Heredero et al, 2021), pharmacological strategies modulating mitochondrial activity may induce exploitable effects on both tumor and immune cells.

Sarcomas are generally considered poorly immunogenic (Fletcher, 2014; Choi & Ro, 2021), but some respond to immunotherapy (D'Angelo et al, 2014; Tawbi et al, 2017; Eulo & Van Tine, 2022). In a recent study, among the 30% complex STSs that presented immune infiltration, the presence of B cell-containing tertiary lymphoid structures (TLS) was associated with the efficacy of anti-PD1 therapy (Petitprez et al, 2020). We reasoned that strategies that exploit the immune-boosting effect of the Vnn1 pathway observed in autoimmune or infectious models (Meghari et al, 2007; Kavian et al, 2016) might enhance the development of anti-sarcoma immunity. Indeed, provision of high VitB5 levels can be obtained in vivo through the administration of pantethine (Pant), a well-tolerated dimeric form of pantetheine hydrolyzed by Vnn1 in VitB5 and cysteamine. In cancer models, in vitro stimulation of CD8$^+$ T cells with CoA could boost the maturation of effector cells with potent antitumor potential (Paul & Ohashi, 2020; St. Paul et al, 2021). Furthermore, melanoma patients with high VitB5 serum levels showed a better response to anti-PD1 therapy (St. Paul et al, 2021). Finally, in an ovarian cancer model, pantethine administration limited tumor growth (Penet et al, 2016). Therefore, we tested whether *VNN1* expression was associated with immune signatures in sarcomas and whether enhancing VitB5 levels via Pant administration could enhance mitochondrial metabolism and anti-tumor immunity in a mouse model.

# Results

### High VNN1 expression in STSs correlates with heightened immune response and better prognosis

We first queried the 224 TCGA STSs samples using the Weighted Gene Correlation Network Analysis (WGCNA) to define the robust gene clusters and to search for those co-expressed with *VNN1*. WGCNA revealed eight gene clusters including a 505-gene cluster containing *VNN1* (Fig S1A). Ontology analysis showed that this cluster was strongly associated with immune MSigDB hallmarks, such as allograft rejection, inflammatory responses or IFNγ response (Fig S1B). To broaden this analysis, we searched for correlations between *VNN1* mRNA expression and clinicopathological and immune variables in an independent cohort of 1,377 clinical STS samples (Fig S1C) collected during surgery for nonmetastatic primary tumors. Their characteristics are summarized in Fig S1D. *VNN1* expression was heterogeneous across the cohort, with a range of intensities over 20 units on the log$_2$ scale (Fig 1A), allowing the search for correlations with tumor variables. *VNN1* expression was assessed as a discrete variable (high versus low) using two different cut-offs on mean or mean ± 0.5 SD. The cut-off based on the strict mean of *VNN1* expression value of the whole series allowed analysis of all samples; VNN1 classes correlated (Fig S1D) with patient's age at diagnosis pathological tumor grade, and tumor site: as compared with the "VNN1-low" class, the "VNN1-high" class included older patients ($P = 3.78 \times 10^{-8}$), more grade 1 tumors ($P = 5.22 \times 10^{-6}$), and more extremity and less superficial trunk tumor sites ($P = 5.20 \times 10^{-3}$). No significant correlation was found between patient's sex, pathological

tumor size, and tumor depth. We then analyzed the correlation between the VNN1 classes and immune variables (Fig 1B). First, the probability of activation of the IFNα, IFNγ, and STAT3 immune pathways was higher in the "VNN1-high" class than in the "VNN1-low" class ($P < 0.05$). Second, we compared the composition and functional orientation of tumor-infiltrated immune cells between VNN1 classes using the Bindea immune cell types defined as the immunome. "VNN1-high" tumors displayed a higher infiltrate concerning 18 immune cell types ($P < 0.05$) including B cells, T cells, effector memory T cells (T$_{em}$), Th1 cells, follicular helper T cells, Th17 cells, CD8$^+$ T cells, γδ T cells, cytotoxic cells, CD56$^{dim}$ NK cells, dendritic cells (DC, interstitial DC, activated DC, and plasmacytoid DC), eosinophils, macrophages, mast cells, and neutrophils. By contrast, "VNN1-low" tumors displayed a higher infiltrate in two immune cell types, NK cells and CD56$^{bright}$ NK cells ($P < 0.05$). Third, "VNN1-high" samples displayed higher Immunologic Constant of Rejection and immune cytolytic activity score than "VNN1-low" samples, reflects of an antitumor cytotoxic immune response; and higher scores for signatures associated with response to immune checkpoint inhibitors (ICI): T-cell-inflamed signature and TLS. Finally, "VNN1-high" samples displayed higher TILs scores using lymphoid-alone, myeloid-alone and combined signatures, and higher antigen-processing molecules score. Next, we searched for correlations between VNN1 classes and clinical outcomes. Patients with "VNN1-high" samples displayed longer MFS than those with "VNN1-low" (Fig 1C with respective 5-yr MFS equal to 66% (95% CI 60–73) versus 61% (95% CI 54–68) (Fig 1D). In the univariate prognostic analysis (Fig 1D), high *VNN1* expression was associated with longer MFS, as was pathological grade 1. In multivariate prognostic analysis, VNN1 status tended towards significance (hazard ratio HR = 0.64, 95% CI 0.41–1.00, $P = 0.051$).

We then applied a second cut-off, more stringent, based on mean ± 0.5 SD, thus decreasing to 754 the number of patients for correlation analyses (343 "VNN1-high-sd" samples and 411 "VNN1-low-sd" samples), including 344 for MFS analysis. Despite this smaller size, we observed similar correlations between the VNN1 classes and the clinicopathological (Fig S2A), immune (Fig S2B), and MFS (Fig S2C and D) data. The patients with "VNN1-high-sd" samples displayed 63% 5-yr MFS (95% CI 55–73), whereas those with "VNN1-low-sd" samples displayed 57% 5-yr MFS (95% CI 47–66) (Fig S2C). In the univariate prognostic analysis (Fig S2D), only *VNN1* expression tended towards significance with longer MFS in the patients with "VNN1-high-sd" samples and a HR for event similar to the one observed with the previous cut-off (HR = 0.70 95% CI 0.48–1.03, $P = 0.072$). Altogether, these results show heterogeneous *VNN1* expression levels in STS, with high *VNN1* levels positively correlated with improved immune infiltration and clinical outcome. Nevertheless, the prognostic value of *VNN1* expression was not observed in a cohort of 326 patients with bone sarcoma, or in a sub-cohort of 94 patients with osteosarcoma (Fig S3A and B).

### Pantethine reduces aggressive fibrosarcoma, melanoma, but not osteosarcoma growth

As a surrogate mouse model of complex sarcomas, we implanted the Vnn1$^{negative}$ MCA 205 (MCA) sarcoma cell line in immunocompetent mice to test whether systemic administration of Pant would compensate for the lack of Vnn1 in the tumor mass. Indeed, pantethine can be hydrolyzed into vitB5 and cysteamine by serum VNN activity (Figs 2A and S4 for the complete CoA synthetic

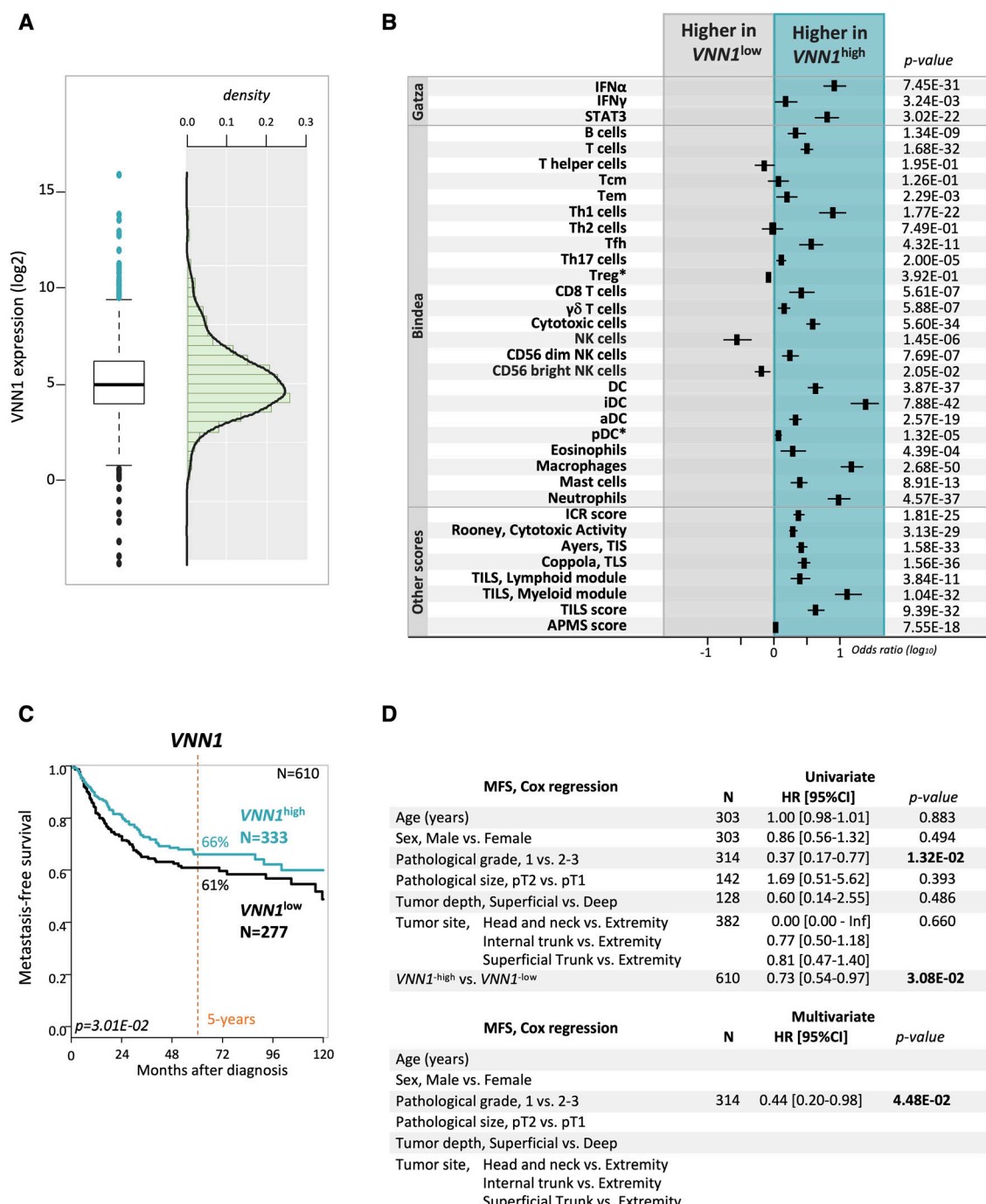

**Figure 1. VNN1 expression has prognostic value in soft-tissue sarcomas.**
**(A)** Box plot showing the distribution of mRNA expression levels of *VNN1* in 1,377 tumor samples. **(B)** Correlations of *VNN1* expression-based classes (high versus low using the mean expression value as cut-off) and immune variables. Forest plots of correlations with the following immune features: immune pathway activation (Gatza), innate and adaptive immune cell subpopulations (Bindea), Immunologic Constant of Rejection score, and cytolytic activity score associated with antitumor cytotoxic immune response, tumor inflammation signature and tertiary lymphoid structure signature associated with response to immune checkpoint inhibitors, and antigen-processing machinery score. The P-values are for the logit link test. N = 1,377, except for two modules indicated by "*" (Treg and pDC) that were informed in 1,152 cases. **(C)** Kaplan–Meier metastasis-free survival curves according to *VNN1* expression (high versus low using the mean expression value as cut-off). The P-value is for the log-rank test. **(D)** Uni and multivariate prognostic analyses for metastasis-free survival. The VNN1 classes are based on the mean expression value as cut-off. The P-values are for the Wald test.
Source data are available for this figure.

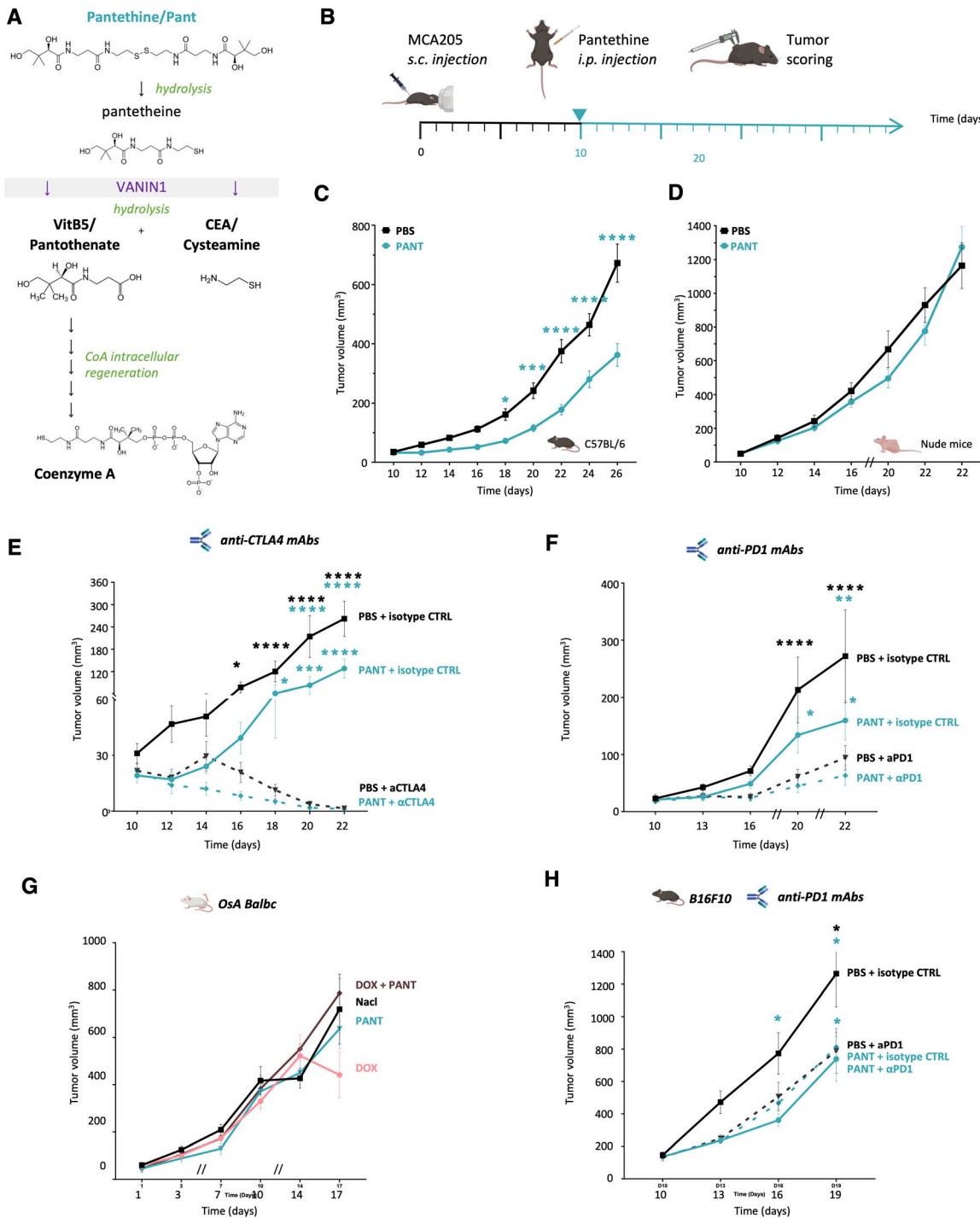

**Figure 2. Pantethine inhibits sarcoma growth.**
**(A)** Enzymatic reaction catalyzed by the Vanin1 pantetheinase: pantetheine is hydrolyzed by Vnn1 into cysteamine and vitamin B5 or pantothenic acid, the latter being the coenzyme A biosynthetic precursor (for the complete pathway of CoA synthesis, refer to Fig S4). **(B)** Schematic representation of the experimental design of murine MCA sarcoma model. **(B, C, D)** MCA205 tumor growth scored after daily injection of Pant in C57BL/6 mice (n = 88) (B) or nude mice (n = 26) (C). Tumor follow-up was interrupted on day 22 in nude mice as limits in authorized tumor sizes were reached. Two-way ANOVA with Šidák's multiple comparisons test. Data are shown with SEM. **(D, E, F)** Effect of Pant therapy in MCA tumor-bearing mice receiving anti-CTLA4 (n = 7 per condition) (D); or anti-PD1 (n = 6 per condition) mAbs (E). **(G)** Growth of OsA orthotopic tumors receiving independent or combined doxorubicine or Pant therapy or vehicle control (n = 6 per condition). **(D, E, H)** Therapeutic effect of Pant therapy, alone (D) or combined with PD1 blockade (E) on B16F10 melanoma growth (n = 10 per condition).
Source data are available for this figure.

pathway). Daily intraperitoneal Pant administration was initiated on established tumors 10 d after cell engraftment (Fig 2B). Growth reduction occurred within a few days post-injection and persisted throughout the treatment period (Fig 2C). A similar result was obtained by co-injecting Pan and cysteamine, the products of pantetheinase activity (Fig S5A) or when using MCA cell line transfected with *Vnn1* as a control of pantethine effect (Fig S5B). Importantly, when Pant was administered to nude mice bearing MCA tumors, no significant growth inhibition was detected (Fig 2D). This result suggested that the Pant effect might directly or indirectly regulate immune responses and possibly modulate the efficacy of ICI. As shown in Fig 2E and F, administration of anti-CTLA4 or anti-PD1 monoclonal antibodies (mAbs) inhibited MCA growth with variable efficacy and combination with Pant modestly accentuated their inhibitory effect. We then tested an osteosarcoma cell line (OsA) for which ICI alone or in combination with doxorubicin had no effect. The addition of Pant treatment did not change the behavior of this highly resistant tumor (Fig 2G) or the organization of the immune infiltrate (Fig S5C and D). Nevertheless, the combined therapy might enhance the extent of necrosis, despite high variability among samples (Fig S5D). In the B16F10 melanoma model, which is known to be sensitive to anti-PD1 therapy (St. Paul et al, 2021), Pant inhibited B16F10 growth to a level comparable with that obtained in MCA tumors and its effect was equivalent to that of an anti-PD1 mAbs treatment or their combined administration (Fig 2H). In conclusion, in the MCA model, Pant efficacy relies on the immunocompetence of the recipient mouse.

## Pantethine boosts myeloid cell signatures associated with antigen presentation and IFN signaling

To further characterize the effect of pantethine therapy on immune responses, we first performed a follow-up of immune cell infiltration at different stages of MCA tumor development. As shown in Fig 3A (see Fig S6A for gating strategy), myeloid cells were approximately twofold more abundant than lymphoid cells in day 13 (D13) tumor infiltrates, with the myeloid (CD11b+)/lymphoid (CD11b−) cell ratio evolving progressively towards an enrichment in lymphocytes after Pant administration. In Pant-treated mice, the proportion of tumor-infiltrating monocytes on D20 was reduced, whereas that of tumor-associated macrophages (TAMs) increased (Fig 3B and C). The latter subset was significantly enriched in MHCII$^{high}$ cells (Fig 3D), previously identified as having potential antitumor functions (Wang et al, 2011).

To further dissect myeloid cell heterogeneity, we performed single-cell RNA sequencing analysis of CD45$^+$ tumor-infiltrating cells at D20 and D28 of tumor progression. We focused our analysis on the three main metaclusters corresponding to myeloid, antigen-presenting cells (APC), and lymphoid cells (Fig S7A). Volcano plot analysis of the myeloid metacluster showed that control tumors were enriched in *Arg1* and *Bnip3*, *Pgam1*, *Ldha*, *Ddit4* transcripts (Figs 3E and S7B for D20 and D28 tumors, respectively) associated with M2 polarization and tumor hypoxia (Semenza, 2013). In contrast, the *MhcII, Cd74, Cxcl9* transcripts were overexpressed in Pant samples. The myeloid (Figs 3F and S7C) and APC (Fig S8A and B) metaclusters were further stratified into subgroups defined by the preferential expression of specific genes (Fig S7D for myeloid cells

and Fig S8B for APC). The density plots distinguished control from Pant-treated samples (Figs S7C and S8C); Pant-treated samples showed progressive accumulation of activated TAMs (identified as mono-TAM B and C and inflammatory TAMs in Fig 3G) and IFNγ-activated DC subsets (Fig 3H). Gene Enrichment Analysis (*GSEA*) (Table 1) and Kyoto Encyclopedia of Genes and Genomes pathways (*KEGG*, Fig S7E) analyses confirmed that Pant samples were enriched in IFNγ-driven transcriptomic profiles associated with antigen presentation (Table 1). APC from Pant tumors expressed high-membrane MHC II levels (Fig S8D). Furthermore, tumor sections documented two to threefold more MHC II$^+$/CD8$^+$ cell contacts (Figs 3I and S8E). To identify the dominant chemokine profiles at the protein level, we performed a proteome analysis (Fig 3J) and a cytometric bead assay (CBA) (Fig S8F). Control PBS samples showed an overrepresentation of proteins associated with tissue clearance (pentraxin), extra cellular matrix (MMP9, Serpin), and endothelial cell reorganization (angiopoietin, VEGF, endostatin). Pant-treated samples contained higher levels of cytokines and chemokines associated with type 1 immunity. This included CCL2 or CXCL9 (Dangaj et al, 2019; Marcovecchio et al, 2021) involved in monocyte (via CCR2) or effector lymphocyte (via CXCR3) recruitment, IL2p40 (Cooper & Khader, 2007; Abdi & Singh, 2015) or Flt3-L (Cueto & Sancho, 2021), associated with M1 or DC maturation (Fig 3J). Within each cluster, we did not observe major differences in the overall expression of genes previously associated with anti or pro-tumor function (Fig S8G). Altogether, Pant samples were enriched in APC subsets, highlighting IFNγ responsiveness, enhanced antigen presentation, and chemoattraction potential.

## Pantethine enhances the development of effector lymphocytes

We used the CD8$^+$/Treg cell ratio as a global indicator of antitumor lymphoid responses and OVA-MCA cells to track tumor-specific CD8$^+$ T cell responses. We observed a progressive enrichment in CD8$^+$ T cells from D17 to D28 (Fig 4A, see Fig S6B for gating strategy). Pant-treated D20 samples contained twofold more NK/ILC, CD4$^+$ in the tumor and OVA-specific effector CD8$^+$ T cells in the tumor-draining lymph nodes (TDLNs) (Fig 4B and C). CD8$^+$ TILs tended to express higher levels of effector molecules such as GZMB, PRF1 or TNFα (Fig 4D). In contrast, the proportion of other immune cells was comparable in lymph nodes or spleen, at steady state or in a tumor context (Fig S6C and D).

The analysis of the lymphoid metacluster (Fig S9A–C) confirmed the cytometry results. On D20, total CD4$^+$ and CD8$^+$ T cells were increased to the expense of Tregs and activated memory CD8$^+$ cells (Figs 4E and S9A–C). CD8$^+$ T cells showed an enriched representation of pathways associated with the IFNγ response, cytotoxic function, chemokine production, and, to a lesser extent, PD1 signaling (Fig 4F and Table 1). On D28, NK/ILC and Treg signatures predominated in Pant and PBS samples, respectively. Furthermore, in Pant-treated samples, twofold fewer CD4$^+$ but not CD8$^+$ T cells expressed CTLA4, PD1 or TIM3 molecules (Fig 4G). We then scored the number of co-expressed genes associated with a cytotoxic program by each cell type. Although this analysis was limited by the total number of cells, CD8$^+$ T cells derived from Pant samples expressed a higher number of cytokine or cytotoxicity genes, an argument in favor of their polyfunctionality (Fig S9D). Finally, we

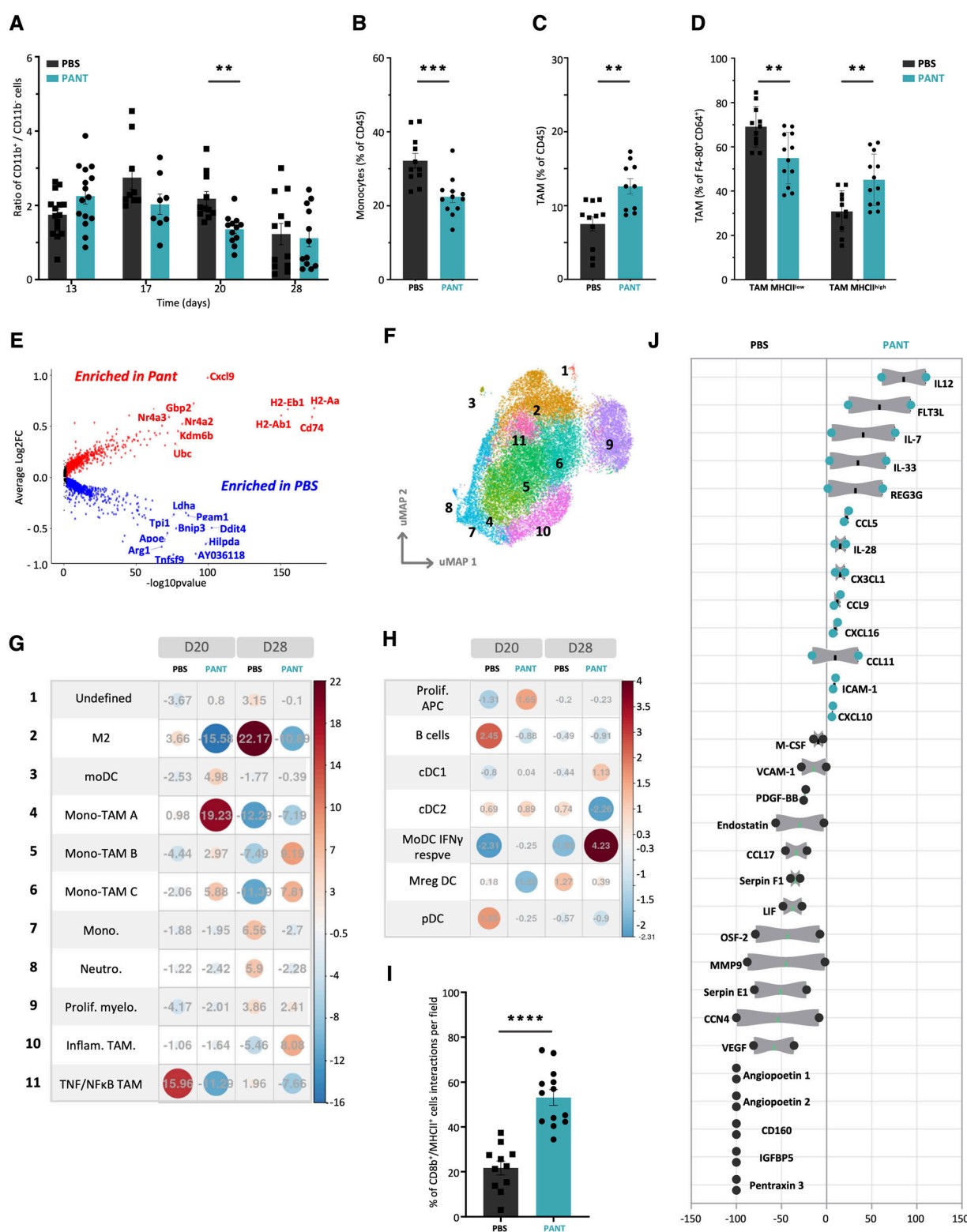

**Figure 3. Analysis of the myeloid compartment.**
**(A)** Kinetic evaluation of the CD11b⁺/CD11b⁻ ratio by flow cytometry analysis of immune infiltrating cells in PBS or Pant tumors (n = 6–8 per condition). Mann–Whitney test; **$P$-value < 0.01. Gating strategies are explained in Fig S6. **(B, C)** Quantification of monocyte (B) and tumor-associated macrophage (TAM) (C) clusters among CD45⁺ cells by flow cytometry on D20 tumors. **(D)** Proportion of MHCII^low and MHCII^high TAMs in total TAM (Ly6C^low F4/80⁺ CD64⁺) PBS and Pant-treated D20 tumors (n = 12). **(E)** Volcano plot representation of myeloid cell transcripts showing differential expression between Pant and PBS samples at D20. Dot colors refer to average log₂FC: positive in red, negative in blue, below the threshold in black. **(A, B, C, F)** Individual uniform manifold approximation, projection, clustering, and population identification

used a global score to recapitulate the expression levels of genes involved in the activation or exhaustion programs in CD8[+−] lymphocytes. Control and Pant samples were comparable for each cell subset, suggesting that Pant administration modifies the proportion, but not the transcriptional state, of these cells (Fig S9E).

Whereas the relative number of total NK/ILC1 was increased in the cytometry analysis, their proportion within the lymphoid metacluster was reduced in the Pant samples. Interestingly, by projecting previously described NK versus ILC1 signatures (Gao et al, 2017; Vienne et al, 2021) (Fig S9F) and applying RNA velocity analysis of the NK/ILC cluster (Fig 4H), we observed an enhanced dynamic conversion of NK into ILC1 cells in control samples, whereas Pant therapy tended to reinforce NK cell signatures, showing by GSEA an enrichment in IFNγ response (Fig S9G). Taken together, our results show that Pant therapy induces a robust effector lymphocyte response.

## Pantethine efficacy depends on IFNγ, cDC1, NK1.1[+], and tumor-infiltrating CD8[+] T cells

The development of cytotoxic CD8[+] T lymphocytes requires antigen cross presentation by XCR1[+] cDC1 cells and tumor-infiltrating potential, a program driven by type 1 cytokines such as IL12 and IFNγ. We tested whether the efficacy of pantethine therapy would persist in the absence of IFNγ signaling and cDC1 cells, using IFNGR1- and *Xcr1*-deficient mice (Wohn et al, 2020). Based on the results presented in Figs 2B and 2D and 5C, the volume of MCA tumors reached 200 mm³ after D20–22. As shown in Fig 5A and B, the lack of Ifngr1 or cDC1 cells led to uncontrollable growth as early as D16. Pant administration had a paradoxically enhancing effect under these conditions. Then, using in vivo depleting anti-NK1.1 or anti-CD8 mAbs, we eliminated cells with cytotoxic potential (Fig S10A and B). As expected, the lack of CD8[+] or NK1.1[+] cells boosted tumor progression after D17 and D22, respectively (Fig 5C and D). Furthermore, Pant-mediated inhibition was preserved in NK1.1[+] cell-depleted mice and was significantly reduced in CD8[+] cell-depleted mice. This showed that the Pant inhibitory effect involved the contribution of both cell types, shown to participate in a DC/NK circuit determining the sensitivity to immunotherapy (Barry et al, 2018). As this cellular pathway is also required for the control of viral infection, we used a model of cutaneous HSV-1 infection (Srivastava et al, 2017) in which virus replication and virus-specific T cells could be monitored. Although the viral load was equivalent under both conditions (Fig 5E), the proportion of central memory and effector CD8[+] T cells in DLNs was enhanced twofold by Pant treatment, including virus-specific CD8[+] T cells (Fig 5F).

Former analyses showed that the levels of CXCL9 and MHC molecules regulated by IFNγ, signaling were enhanced by Pant therapy (Table 1). To test whether Pant might influence antigen presentation or T cell proliferation, we injected OVA-expressing MCA cells, extracted CD11c[+] DC from TDLNs and co-cultured them in vitro with CFSE-labeled OT-1 CD8[+] T cells in the presence or absence of OVA. Both dendritic cell numbers (Fig S10C) and CD8[+] T cell proliferation index (Fig S10D) were comparable under the two conditions. We then performed immunohistochemical analysis in tumor sections that showed a higher proportion of CXCL9-dependent infiltrating CD3[+] and CD8[+] TILs (Chow et al, 2019; Dangaj et al, 2019; Marcovecchio et al, 2021) in Pant samples (Fig 5G and H), whereas they were in reduced numbers or confined to the periphery in control tumors. Altogether, these results confirm the boosting effect of Pant therapy on IFNγ-regulated pathways.

## Pantethine therapy enhances VitB5-dependent metabolism in tumor T lymphocytes

Vnn1 pantetheinase hydrolyzes Pant in VitB5, which supports CoA synthesis and several aspects of lipid metabolism (Naquet et al, 2020). We first confirmed by metabolomics analysis that metabolites linked to pantothenate and redox (glutathione, taurine) metabolism were enriched in tumor masses after Pant therapy (Figs 6A and S11). This analysis also documented a reduction in several fatty acids and inflammatory lipids linked to the lipid oxidation pathway (Fig S11), whereas the metabolism of phospholipids was augmented. Because prolonged Pant administration might progressively induce metabolic adaptations in tissues, we performed another analysis on D28 tumors and compared the Pant effect to that induced by dichloroacetate (DCA), a compound that increases pyruvate uptake in mitochondria via PDK inhibition (Michelakis et al, 2008) and exerts comparable but not synergistic antitumor activity on MCA cells (Fig 6B) and other tumors (Tataranni & Piccoli, 2019). The results showed that global metabolic changes were not superimposable (Fig 6C). Pant enhanced the levels of metabolites associated with the stress response, and protein and nucleotide metabolisms, whereas DCA enhanced carbohydrate metabolism and mitochondrial activity. Interestingly, their association had a significant effect on the production of mitochondrial energetic metabolites and biogenic amines, a phenotype confirmed by nuclear magnetic resonance (NMR) analysis (Fig S12A).

We then focused our metabolic analysis on in vitro grown autochthonous or ex vivo isolated cancer cells. In vitro, addition of Pant slightly enhanced the production of mitochondrial reactive oxygen species (mtROS) (Fig 6D). In contrast, the oxygen consumption (OCR) and extracellular acidification rates (ECAR) measured by Seahorse analysis were marginally affected (Fig 6E), showing that these tumor cells have a limited spared respiratory capacity. Because chronic stress may progressively induce mitochondrial damage/repair processes in tumor cells, we comparatively evaluated mitochondrial fitness in vivo, by scoring using flow cytometry from D24 to D28, the mitochondrial polarization over mass ratio (MDR/MG, see the Materials and Methods section) (Elefantova et al, 2018). As shown in Figs 6F and S12B, depolarized

---

computed for the myeloid metacluster delineated 11 clusters: neutrophils, monocytes, Mono-TAM (A, B, C), inflammatory TAM, TAM TNFα/NFκB, moDC, CAF, M2, and Ki67 TAM. **(G, H)** Pearson's residuals of TAM and APC cell clusters from single-cell dataset obtained from CD45-enriched cells from tumors harvested at day 20 and D28 of PBS or Pant-treated mice (n = 2). Cell subset enrichment is represented on a colored scale from red (enriched in PANT) to blue (enriched in PBS). **(I)** Quantification of cell contacts between CD8β[+] and MHC II[+] cells on D24 tumor sections shown in Fig S7E. Mann–Whitney test; ****P-value < 0.0001. **(J)** Proteomic profiling of tumor lysates represented by enrichment in Pant versus PBS condition (n = 4).
Source data are available for this figure.

**Table 1. Enriched pathways in cell subsets.**

| | D20 | | | | D28 | | | |
|---|---|---|---|---|---|---|---|---|
| | **Pathway** | **NES** | ***Pval*** | ***Padj*** | **Pathway** | **NES** | ***Pval*** | ***Padj*** |
| Monocytes | IFNg response | 2.12 | $1.00 \times 10^{-7}$ | $2.50 \times 10^{-6}$ | IFNg response | 2.9 | $7.70 \times 10^{-24}$ | $1.90 \times 10^{-22}$ |
| | Ag processing and pres. | 1.75 | $2.00 \times 10^{-3}$ | $1.20 \times 10^{-2}$ | Ag processing and pres. | 2.47 | $1.30 \times 10^{-10}$ | $1.70 \times 10^{-9}$ |
| | OXPHOS | −1.85 | $3.40 \times 10^{-5}$ | $3.60 \times 10^{-4}$ | OXPHOS | −1.96 | $6.10 \times 10^{-6}$ | $3.80 \times 10^{-5}$ |
| | Aerobic respiration | −1.5 | $1.00 \times 10^{-2}$ | $3.60 \times 10^{-2}$ | | | | |
| Mono-TAM1 | IFNg response | 2.42 | $5.00 \times 10^{-12}$ | $1.20 \times 10^{-10}$ | IFNg response | 2.76 | $4.20 \times 10^{-19}$ | $1.10 \times 10^{-17}$ |
| | Ag processing and pres. | 1.8 | $9.10 \times 10^{-4}$ | $7.60 \times 10^{-3}$ | Ag processing and pres. | 2.77 | $2.10 \times 10^{-14}$ | $2.60 \times 10^{-13}$ |
| | | | | | Aerobic respiration | −2.32 | $9.20 \times 10^{-10}$ | $7.70 \times 10^{-9}$ |
| | | | | | OXPHOS | −1.79 | $4.90 \times 10^{-5}$ | $3.00 \times 10^{-4}$ |
| Mono-TAM3 | IFNg response | 1.57 | $4.20 \times 10^{-3}$ | $3.50 \times 10^{-2}$ | IFNg response | 2.46 | $1.10 \times 10^{-12}$ | $8.80 \times 10^{-12}$ |
| | | | | | Ag processing and pres. | 2.58 | $1.20 \times 10^{-11}$ | $7.50 \times 10^{-11}$ |
| | | | | | Aerobic respiration | −2.81 | $1.80 \times 10^{-15}$ | $4.40 \times 10^{-14}$ |
| | | | | | OXPHOS | −2.57 | $1.80 \times 10^{-13}$ | $2.20 \times 10^{-12}$ |
| TAM Inf | Ag processing and pres. | 1.72 | $1.60 \times 10^{-3}$ | $2.00 \times 10^{-2}$ | Ag processing and pres. | 2.34 | $4.90 \times 10^{-10}$ | $6.90 \times 10^{-9}$ |
| | | | | | OXPHOS | −2.26 | $1.50 \times 10^{-9}$ | $1.20 \times 10^{-8}$ |
| | | | | | Aerobic respiration | −2.86 | $6.80 \times 10^{-17}$ | $1.70 \times 10^{-15}$ |
| TAM TNF NF-kB | Chemokine production | 1.67 | $5.10 \times 10^{-3}$ | $3.20 \times 10^{-2}$ | IFNg response | 1.82 | $7.10 \times 10^{-5}$ | $5.90 \times 10^{-4}$ |
| | OXPHOS | −1.79 | $7.10 \times 10^{-6}$ | $1.80 \times 10^{-4}$ | OXPHOS | −2.36 | $9.50 \times 10^{-12}$ | $2.10 \times 10^{-10}$ |
| | Aerobic respiration | −1.56 | $8.90 \times 10^{-4}$ | $1.10 \times 10^{-2}$ | Aerobic respiration | −1.96 | $5.40 \times 10^{-6}$ | $6.70 \times 10^{-5}$ |
| M2-like TAM | | | | | IFNg response | 2.74 | $9.20 \times 10^{-15}$ | $2.30 \times 10^{-13}$ |
| | | | | | OXPHOS | −2.42 | $1.50 \times 10^{-12}$ | $1.90 \times 10^{-11}$ |
| | | | | | Aerobic respiration | −2.47 | $9.10 \times 10^{-12}$ | $7.50 \times 10^{-11}$ |
| APC | IFNg response | 2.48 | $8.50 \times 10^{-12}$ | $2.10 \times 10^{-10}$ | IFNg response | 2.47 | $4.90 \times 10^{-11}$ | $1.20 \times 10^{-9}$ |
| | Ag processing and pres. | 1.91 | $4.50 \times 10^{-4}$ | $5.60 \times 10^{-3}$ | Ag processing and pres. | 1.73 | $2.70 \times 10^{-3}$ | $3.30 \times 10^{-2}$ |
| CD8 T | IFNg response | −2.32 | $1.50 \times 10^{-8}$ | $3.70 \times 10^{-7}$ | IFNg response | 1.62 | $1.70 \times 10^{-3}$ | $4.20 \times 10^{-2}$ |
| Proliferating CD8 | IFNg response | 2.04 | $2.00 \times 10^{-6}$ | $5.10 \times 10^{-5}$ | IFNg response | 1.86 | $1.40 \times 10^{-6}$ | $1.30 \times 10^{-3}$ |
| | Chemokine production | 2.07 | $9.70 \times 10^{-5}$ | $1.20 \times 10^{-3}$ | PD1 signaling | 2.02 | $1.00 \times 10^{-4}$ | $3.10 \times 10^{-3}$ |
| | Cytotoxic pathway | 2 | $3.70 \times 10^{-4}$ | $3.10 \times 10^{-3}$ | | | | |
| | PD1 signaling | 1.83 | $4.50 \times 10^{-3}$ | $2.20 \times 10^{-2}$ | | | | |
| | OXPHOS | −1.51 | $2.60 \times 10^{-3}$ | $1.60 \times 10^{-2}$ | | | | |

mitochondria accumulated in D28 tumors in favor of increased stress, and the MDR/MG ratio evolved similarly between PBS and Pant samples. Because alterations in mitochondrial metabolism can contribute to the reinforcement of the Warburg effect (Senyilmaz & Teleman, 2015; Liberti & Locasale, 2016), we tested lactate production in tumor masses (Fig 6G) and quantified by qRT–PCR the expression level of transcripts associated with an increased glycolytic flow on isolated cancer cells (Fig S12C). The results showed that neither lactate nor transcript levels were modified by Pant. Altogether, Pant administration does not seem to affect mitochondrial and glycolytic activities by MCA cells.

Consequently, immune cells may be the preferential targets of Pant therapy in this model. In purified $CD8^+$ lymphocytes extracted from D20 tumors, Pant administration provoked a significant increase in both OCR and ECAR, demonstrating their higher energy statuses (Fig 6H). To follow mitochondrial fitness over time, we quantified mitochondrial polarization and mass and deduced the MDR/MG ratio in $CD4^+$ and $CD8^+$ T cells at different time points. Although no difference was observed in both cell subsets on D20, this ratio was lowered in D28 $CD4^+$ and $CD8^+$ T cells from the Pant samples (Fig 6I). These results showed that Pant boosts mitochondrial metabolism in vivo but does not prevent the development of mitochondrial depolarization (Table 1) associated with chronic lymphocyte activation and exhaustion (Yu et al, 2020).

## Discussion

Pantethine administration that aimed at restoring VitB5 and mitochondrial homeostasis, led to the reduced growth of an

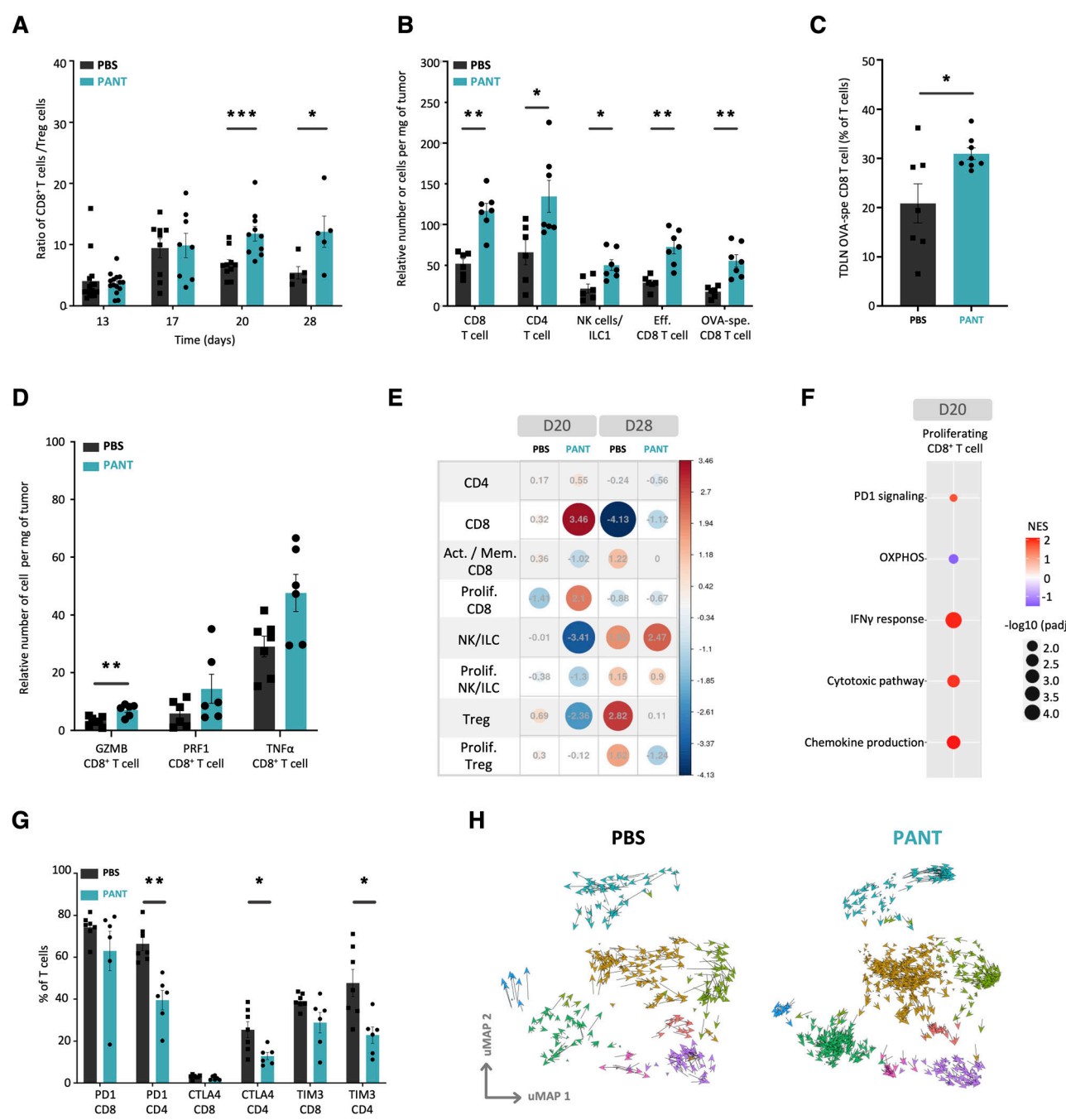

**Figure 4. Analysis of the lymphoid compartment.**
**(A)** Kinetic evaluation of the CD8+/CD4+ Treg cell ratio by flow cytometry among immune infiltrated cells in tumor from PBS or Pant samples (n = 5 per condition). Mann–Whitney test; * *P*-value < 0.05. Gating strategy shown in Fig S8B. **(B)** Relative numbers of tumor-infiltrating lymphocytes per mg of PBS and Pant tumors. **(C)** Quantification of OVA-specific CD8 T cells in tumor draining lymph node evaluated by flow cytometry using a SIINFEKL tetramer. **(C, D)** Flow cytometry analysis of effector molecule expression by CD8 T cells isolated from day 20 tumors and in vitro restimulated in the presence of PMA/ionomycin/brefeldin A (n = 6) (C). Mann–Whitney test; **P*-value < 0.01; * *P*-value < 0.05. **(E)** Pearson's residuals of lymphoid cell clusters from a single-cell dataset obtained from CD45-enriched cells from tumors harvested at day 20 and D28 of PBS or Pant samples (n = 2). **(F)** Graphic representation of Gene Set Enrichment Analyses performed in the proliferating CD8 T cell subset as indicated in Table 1. **(G)** Flow cytometry analysis of immune checkpoint expression by tumor-infiltrating lymphocytes from PBS and Pant-treated mice at day 28 (n = 6). Mann–Whitney test; **P*-value < 0.01; * *P*-value < 0.05. **(H)** Projection of individual cell velocity as determined by scVelo analysis performed on each subset of lymphoid cells from D20 samples.
Source data are available for this figure.

aggressive sarcoma cell line in immunocompetent mice. Pant acts predominantly via cysteamine-dependent redox modulation and CoA regeneration. An intrinsic tumor-suppressive effect via

metabolic rewiring was unlikely in the MCA model, which contradicts a previous report in which the mitochondrial activity of H1 cells could be rescued by exogenous VitB5 (Giessner et al, 2018). In

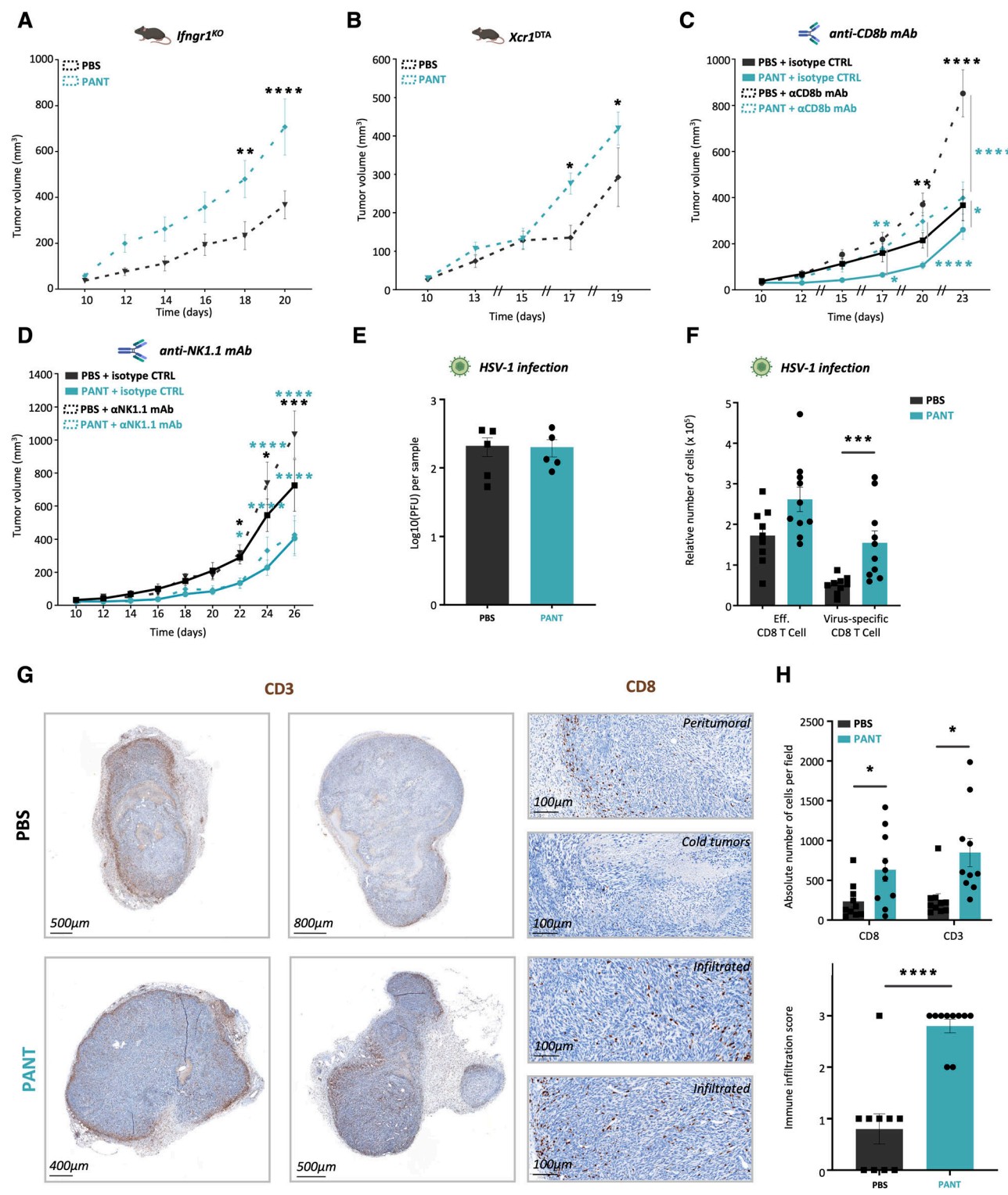

**Figure 5. Requirements for Pant antitumor efficacy.**
**(A, B, C, D, E, F)** MCA growth curves in *Ifnrgr1*-deficient (n = 7–8 per condition) (A), cDC1 cell-deficient *XCR1*^DTA (n = 10–11 per condition) (E), anti CD8⁻ (n = 10–11 per condition) or anti NK1.1-treated (n = 8 per condition) (F) control or Pant-treated mice. Two-way ANOVA with Šidák's multiple comparisons test; ****$P$-value < 0.0001, **$P$-value < 0.01, * $P$-value < 0.05. **(E, F, G, H)** PFU quantifying viral load (G) and numbers of CD8⁺ T cell subpopulations (H) in the draining lymph nodes of HSV1-infected mice after 8 d of PBS or Pant treatment. **(G)** Immunochemistry analysis of PBS and Pant tumor sections using anti-CD3 and anti-CD8 mAbs on D24. **(H)** Quantification of the absolute number of positive cells per field (top panel) and of the immune infiltration score (bottom panel). n = 10. Mann–Whitney test; ****$P$-value < 0.0001, * $P$-value < 0.05.
Source data are available for this figure.

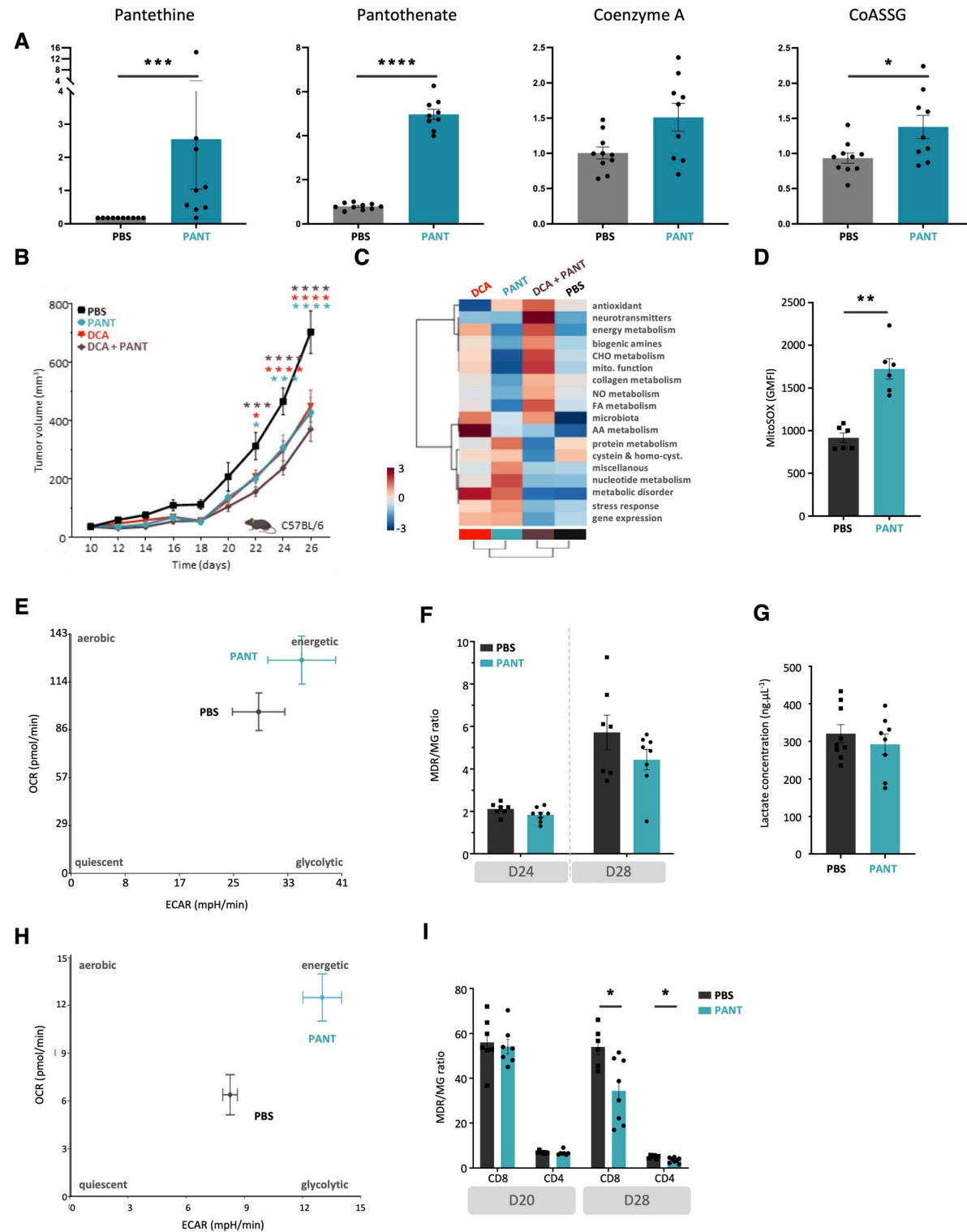

**Figure 6. Metabolic changes induced by Pant administration.**
**(A)** Metabolomics quantification of VitB5-related metabolites with or without Pant administration in tumor masses (n = 7 per condition). **(B)** MCA205 tumor growth scored after daily injections of Pant, dichloroacetate (DCA) combined with Pant or not in C57BL/6 mice (n = 24). Two-way ANOVA with Šídák's multiple comparisons test; ****$P$-value < 0.0001, ***$P$-value < 0.001, **$P$-value < 0.01, * $P$-value < 0.05. Data are shown with SEM. **(C)** Metabolomics analysis of total tumors from mice treated with PBS, Pant, DCA, Pant + DCA by LC-MS (n = 10) showing the hierarchical cluster of metabolic pathways according to treatment conditions. Scaling is in the unit of variance (mean/squared root of SD). **(D)** Geometric mean fluorescence intensity of MitoSox-stained MCA cells incubated or not for 4 h in Pant-enriched medium in vitro (n = 6 per condition). Mann–Whitney test; **$P$-value < 0.01. **(E)** Seahorse analysis of an energetic map integrating oxygen consumption and extracellular acidification rates measurements of in vitro Pant-treated MCA205 or PBS control cells (n = 2). **(F)** MDR/MG ratio of CD45⁻ cells at D24 and D28 from Pant-treated or PBS control mice (n = 7). **(G)** Lactate concentration quantified in whole-tumor

contrast, MCA cells originate from MCA-driven tumorigenesis, with a long evolution time in vivo (Vienne et al, 2021). As many aggressive tumors display mitochondrial alterations (Miallot et al, 2021), MCA cells show reduced respiratory capacity and, consequently, Pant neither corrects mitochondrial fitness nor inhibits tumor growth in nude mice. It rather enhanced mtROS generation, a phenotype observed in cells with electron transport chain deficiencies (Zorov et al, 2014), and envisaged as a therapeutic strategy for embryonal rhabdomyosarcoma (Chen et al, 2013) to induce cell death or immunostimulation via damage-associated molecular pattern release (Porporato, 2018; Rongvaux, 2018). Therefore, we suspected that metabolic changes induced by Pant could boost antitumor immunity. In Pant-treated mice, tumor-associated myeloid and lymphoid cells displayed signatures of IFNγ-driven activation. The contribution of mitochondrial activity to antigen recognition and the efficacy of immunotherapy is still debated (Bonifaz et al, 2014; Harel et al, 2019; Jones & Divakaruni, 2020; Yin & O'Neill, 2021). In macrophages, CoA administration enhances acetyl-CoA–mediated epigenetic activation of pro-inflammatory gene transcription and boosts TAM antitumor activity in breast cancer (Timblin et al, 2022 Preprint). Although dendritic cells require enhanced mitochondrial activity for the acquisition of antigen-presenting capacity in vitro, we failed to detect any Pant-mediated enhancement in OVA presentation to OT1 T cells by cDC1 extracted from Pant-treated mice. In vivo, Pant therapeutic efficacy requires the presence of IFNγ signaling, cDC1, and NK1[+] and CD8[+] TILS. Indeed, cDC1 cells cross present antigens (Dorner et al, 2009; Broz et al, 2014; Brewitz et al, 2017) and, through cooperation with Th1 cytokine-producing NK or CD4[+] Th1 cells (Barry et al, 2018; Wohn et al, 2020), prime antitumor CD8[+] T cell responses. Interestingly, perturbation of fatty acid oxidation or mitochondrial clearance increases the development of cDC2 at the expense of cDC1 cells (Kratchmarov et al, 2018). Furthermore, CD8[+] DC boosts mitochondrial energy production to maintain cross presentation or IL-12 production (Farrand et al, 2009; Joshi & Pillai, 2018; Oberkampf et al, 2018). Interestingly, tumors from Pant-treated mice showed increased IL12p40 and CXCL9 production involved in Th1 effector cell priming and recruitment via the CXCR3 axis (Chow et al, 2019). Mitochondrial activity is also essential for T lymphocyte activation, cytokine production, and differentiation (van der Windt & Pearce, 2012; Chang et al, 2013; Sena et al, 2013; Almeida et al, 2021), in part through mtROS (Kamiński et al, 2012; Sena et al, 2013; Phan et al, 2016; Ma et al, 2019). Accordingly, they were two–threefold more infiltrated by CD4[+] or CD8[+] effector T cells, in which the simultaneous expression of Th1 cytokines and cytolytic activity markers was enhanced, an argument in favor of their polyfunctionality. In the D20 Pant-treated samples, CD8[+] T cells showed a hypermetabolic profile with simultaneous enhancement of ECAR and OCR. Accordingly, TLR-induced IFNγ production by CD8[+] T cells depends on mitochondrial ATP (Salerno et al, 2016). Electron transport chain complexes 1 and 2 are differentially required for Th1 cell proliferation and epigenetic

remodeling versus the maintenance of terminal effector functions, respectively (Bailis et al, 2019). Interestingly, velocity index analyses suggested that the identity of the effector innate or adaptive lymphocytes was reinforced by Pant therapy. Our results indicate that Pant, through its effect on mitochondrial fitness, may enhance the functionality of antitumor cells and/or their ability to migrate within the tumor mass. They extended recent findings showing that, in vitro, addition of CoA could enhance the production of effector memory CD8[+] T cells that upon in vivo injection rejected a tumor (St. Paul et al, 2021). In this study, pantothenate administration also enhanced the efficacy of anti-PDL1 therapy in the MC38 model, a situation that was at the limit of detection using the MCA and B16 models treated with anti-PD1 or CTLA4 therapy.

Several explanations may explain the incomplete effect of Pant administration. First, pro-tumor cells may also be activated by Pant. Accordingly, in the absence of cDC1 or IFNγ signaling, Pant enhanced tumor growth. Indeed, CoA and its acylated forms have a strong impact on fatty acid oxidation and mitochondrial activity that participate in pro-tumorigenic type 2 immunity through the regulation of macrophage (Jha et al, 2015) and Treg functions (Field et al, 2020). Second, chronic activation of lymphocytes commonly leads to cell exhaustion (Blank et al, 2019; Yang et al, 2020) and expression of inhibitory PD1 or TIM3 molecules. This phenotype is correlated with long-lived effector cells that survive the contraction phase of the immune response, but develop various dysfunctions and mitochondrial alterations (Scharping et al, 2016; Blank et al, 2019), in part, induced by inhibitory molecules such as PD1 (Bengsch et al, 2016; Chowdhury et al, 2018; Wei et al, 2018). Our results indicated that PD1/TIM3 expression was lower in CD4[+] but not CD8[+] T cells from Pant-treated tumors than in control situation. Furthermore, mitochondrial depolarization was more important in CD8[+] T cells from Pant-treated mice, likely because of the higher activation level at various stages of tumor development. Although the difference between the control and Pant samples was not detected at the transcriptional level for each cell subset, the progressive loss of functionality might limit Pant efficiency and explain the lack of synergy with anti-PD1 therapy. Indeed, Pant and anti-PD1 therapies were equally effective or ineffective on B16 melanoma or osteosarcoma cell lines, respectively, possibly because they converge on similar metabolic pathways (Chamoto et al, 2017).

Pant administration mimics the biological effects linked to the overexpression of Vnn1 pantetheinase in vivo (Giessner et al, 2018; Millet et al, 2022). More generally, Vnn1 is involved in tissue tolerance to stress (Naquet et al, 2014). Therefore, we tested whether *VNN1* expression in human sarcomas might be correlated with enhanced survival and/or development of antitumor responses. We analyzed large retrospective series of STSs (n = 1,377) and bone sarcomas (n = 326). *VNN1* expression was variable in STSs and inversely correlated with their severity. However, when detected and using two different cut-offs, its expression positively correlated with improved prognosis and the presence of immune signatures

masses from PBS or Pant-treated mice at day 20 post engraftment (n = 8 per condition). **(H)** Energetic map of CD8[+] tumor-infiltrating lymphocytes isolated from tumor mass D20 post cell engraftment. Mice were treated with PBS or Pant starting on day 10 (n = 3). **(I)** MDR/MG ratio of CD8[+] and CD4[+] tumor-infiltrating lymphocytes at D20 and D28 from Pant-treated or PBS control mice (n = 7). Mann–Whitney test; * P-value < 0.05.
Source data are available for this figure.

recapitulating the phenotypes observed in the mouse model. Conversely, in human bone sarcomas, *VNN1* expression does not correlate with prognosis, and Pant does not control tumor growth in mice. This phenotype may be related to the fact that loss of mtDNA is often found associated with aggressive osteosarcomas (Jackson et al, 2019), whereas its enhanced release can boost cDC1-dependent immune responses in MCA tumors (Miallot et al, 2023a). Such in silico analyses display strengths and limitations. The first limitation is related to the retrospective nature and its associated biases such as heterogeneity and missing data. For example, survival data were available for 610 STS patients and age for 532 patients. However, to the best of our knowledge, our series is the largest one analyzing *VNN1* expression in STSs. Other strengths include its originality, being the first one to describe the clinico-pathological and molecular correlations with *VNN1* expression in STS samples, and the use of a patients' population treated in the community setting, likely more reflective of real life than a selected patient population within a clinical trial. Finally, our results expand the links between the efficacy of the CoA biosynthetic pathway and cancer prognosis. Few studies already suggested that the levels of PANK1 and 2, but not 4, isoforms, known to positively regulate CoA rate of synthesis (Giessner et al, 2018), were positively associated with a better outcome in clear cell renal cell carcinoma and acute myeloid leukemia, respectively (Liu et al, 2019; Wang et al, 2022).

In conclusion, our results extend our former observation that VitB5 is a limiting metabolite in many tumors, and that this defect participates in the impairment of anti-tumor immunity. Pan-tetheine may present a therapeutic interest in the context of immune enhancement before surgery or in vaccine programs.

# Materials and Methods

### Animals

Female 8–10 wk old C57BL/6 and nude mice were purchased from Janvier Laboratories. *Xcr1*[DTA] and OT-I (ovalbumin-specific TCR-transgenic mice) mice obtained from B Malissen's laboratory have been previously described (Voehringer et al, 2008; Wohn et al, 2020). *Ifngr1*[KO] mice were purchased from CNRS (UMR 7355). Mice were housed under a standard 12-h:12-h light–dark cycle with ad libitum access to food and nonacid water, 22°C ±1°C, 45–60% humidity, and maintained under specific pathogen-free conditions at an animal facility of the CIML (F13-055-10). For the osteosarcoma murine model, 5-wk-old BALB/cByj mice were purchased from Charles River Laboratories and housed at CRCL animal facilities under the same conditions as mentioned previously. All experiments were conducted in accordance with institutional committees and the French and European guidelines for animal care. The experiments were approved by the Ethical Committee for Animal Experimentation.

### Cell lines

The MCA205 mouse fibrosarcoma cell line (SCC173), the OVA-expressing cells (a kind gift from Laurence Zitvogel, Institut Gustave Roussy, Villejuif, France), and the B16F10 melanoma cell line (ATCC-CRL-6475) were cultured in DMEM F-12 (Thermo Fisher Scientific) supplemented with 10% FBS (Gibco), 100 units/ml penicillin–streptomycin (Gibco), 2 mM of L-glutamine, and 1 mM of sodium pyruvate at 37°C with 10% $CO_2$. Murine OsA K7M2 (CRl-2836; ATCC) was cultured in complete DMEM and maintained for 2 wk before mouse cell engraftment. Mycoplasma status was evaluated using the MycoAlert Lonza Detection kit and cells were used at low passage.

### Treatments

For drug treatment mice were injected intraperitoneally (*i.p.*) starting at day 10 post cell engraftment. D-pantethine (1 g/Kg, 45 µmol per injection per mouse) was administered every day and DCA (1 g/Kg) every 2 d. Cysteamine and pantothenate were administered *i.p.* at 120 mg/kg (6 µmol per injection per mouse, toxic beyond this dose) and 500 mg/kg (50 µmol per injection per mouse), respectively. Concentrated stocks were prepared in water and diluted in PBS for injection. For all experiments, control PBS vehicle was used. All reagents were purchased from Sigma-Aldrich. For osteosarcoma experiment, mice were treated with doxorubicine (0.75 mg/Kg; Baxter) or NaCl vehicle 2 d a week. The anti-CTLA4 (clone 9H10, 100 µg/injection/mouse; BioXcell), anti-PD1 antibodies (clone RMP1-14, 200 µg/injection/mouse; BioXcell), and isotype control mAbs (InVivoMab polyclonal Syrian hamster IgG and InVivoMab rat IgG2a, clone2A3; BioXcell) were injected four times at 3-d intervals *i.p.*

### Tumor experiments

A total of $10^5$ MCA 205 cells or OVA-expressing MCA205 were subcutaneously and bilaterally grafted into the flanks of C57BL/6, nude mice, *XCR1*[DTA] mice, and *Ifngr1*[KO] mice. $3.10^5$ B16F10 were injected intradermally and bilaterally in C57BL/6 mice. Murine OsA K7M2 was injected into the tibia of BALB/cByj mice. Mice were anesthetized with 2.5% isoflurane every 2 d and tumor growth was scored using a caliper by measuring the length (L) and width (W) of the tumor and volumes were calculated using the following formula: $(L × W)^2/2$. A limit point was set when the tumor volume was >1,500 $mm^3$ or when symptoms of poor health were detected (weight loss, prostration, bleeding, and scars). Tumors were harvested between D10–D28 post cell engraftment, and each tumor was considered as an experimental unit referred to as *n*. Tumors were mechanically and enzymatically digested using Miltenyi Biotech GentleMACS Octo Dissociator technology. The samples were filtered through a 70-µm cell strainer (Becton Dickinson) and subjected to red blood cells lysis (eBioscience). The cell suspensions were analyzed using flow cytometry. Alternatively, CD45$^-$ cells, CD8a$^+$ naïve T cells, and CD8$^+$ TILs were isolated using microbeads from Miltenyi Biotech and MultiMACS Cell24 separation on LS columns according to the manufacturer's protocols. The inguinal TDLNs were collected between D14 and D28 post-cell engraftment, scratched on a 70-µm cell strainer and analyzed by flow cytometry. Spleen samples were processed as described for the lymph nodes. For osteosarcoma experiments, the tumors were harvested, fixed in PFA 4%, included in paraffin for subsequent histological analysis.

**LC–MS**

Two independent metabolomics analyses were performed from tumor samples. Metabolomics experiments to detect VitB5-related metabolites were performed by Metabolon (the complete metabolite data are available in Supplemental Data 1). In this experiment, nine samples were obtained from pantothenate-treated mice and 10 from control tumors. To remove batch variability, for each metabolite, the values in the experimental samples are divided by the median of those samples in each instrument batch, giving each batch, and thus the metabolite, a median of one. For each metabolite, the minimum value across all batches in the median-scaled data is imputed for the missing values. The Batch-norm-Imputed Data are transformed using the natural log. Metabolomics data typically display a log-normal distribution; therefore, the log-transformed data are used for statistical analyses. Pantethine, pantothenate, CoA, and CoA-SSG are detected in negative mode. For all metabolomics analyses, all dried polar extracts from the samples were reconstituted with 200 $\mu$l of acetonitrile/water (50:50; v:v). The samples were separated using high-performance liquid chromatography (UPLC) Ultimate 3000 (Thermo Fisher Scientific), coupled to a high-resolution mass spectrometer. The Q-Exactive Plus quadrupole-orbitrap hybrid was equipped with an electrospray ionization source (H-ESI II). The chromatographic separation was performed on a binary solvent system using a reverse phase C18 column (Hypersil Gold, 100 mm × 2.1 mm, 1.9 $\mu$m; Thermo Fisher Scientific) at 40°C with a flow rate of 0.4 ml min$^{-1}$ and a HILIC column (150 mm × 2.1 mm, 5 $\mu$m, 200 A; Merk, SeQuant ZIC-HILIC) at 25°C with a flow rate of 0.25 ml min$^{-1}$. The injection volume for both columns was 5 $\mu$l. The chromatographic separation was performed on a binary solvent system at a flow rate of 0.4 ml min$^{-1}$ using a reverse phase C18 column (Hypersil Gold, 100 mm × 2.1 mm, 1.9 $\mu$m; Thermo Fisher Scientific) at 40°C. The mobile phase consisted of a combination of solvents A (0.1% formic acid in water, vol/vol) and B (0.1% formic acid in acetonitrile, vol/vol). The injection volume was 5 $\mu$l. The following gradient conditions were used: 0–1 min, isocratic 100% A; 1–11 min, linear from 0% to 100% B; 11–13 min, isocratic 100% B; 13–14 min, linear from 100% to 0% B; 14–16 min, isocratic 100% A. The separated molecules were analyzed in both the positive and negative ionization modes in the same run. The mass spectra were collected using a resolving power 35,000 full width at half maximum for the theoretical mass-to-charge ratio (m/z) of 200. Full-scan mass spectra were acquired in the m/z range of 80–1,000. The ionization source parameters for positive and negative ion modes were as follows: capillary temperature 320°C, spray voltage 3.5 kV, sheath gas 30 (arbitrary units), auxiliary gas eight (arbitrary units), probe heater temperature 310°C, and S-Lens RF level was set at 55 V. MS/MS experiments were performed using higher energy collision-induced dissociation (HCD) and the normalized collision energy applied was ramped from 10% to 40%. To ensure good repeatability of the analysis, a quality control sample (QC) was formed by pooling a small aliquot of each biological sample. The QC sample was analyzed intermittently (because of the small number of samples, QC was analyzed one out of every three samples) for the duration of the analytical study to assess the variance observed in the data throughout the sample preparation, data acquisition, and data pre-processing steps. The mobile phase for HILIC column separation consisted of a combination of solvent A (100% water, 16 mM ammonium formate) and solvent B (100% acetonitrile 0.1% formic acid). The following gradient conditions were used: 0–2 min, isocratic 97% B; 2–10 min, linear from 97% to 70% B;

10–15 min, linear 70–10% B; 15–17 min, isocratic 10% B; 17–18 min linear from 10% to 97% B; from 18 to 22 min isocratic 97% B. The separated molecules were analyzed in both the positive and negative ionization modes in the same run. The mass acquisition parameters were the same as those used for the C18 columns. The repeatability of the analysis was ensured by analyzing the quality control sample (QC) intermittently (one out of every five samples).

**NMR**

Samples of the thawed tissue (~15 mg) were placed into a 30-$\mu$l disposable insert, in which 10 $\mu$l of D2O was added to provide a field-lock signal. The disposable insert was then inserted into a 4-mm ZrO2 HRMAS rotor for HRMAS NMR spectral acquisition. All NMR experiments were carried out on a Bruker Avance III spectrometer operating at 400 MHz for the 1H frequency equipped with a 1H/13C/31P HRMAS probe. Spectra were obtained at 278 K with a spin rate of 4 kHz. To attenuate the NMR signals of macromolecules, the Carr–Purcell–Meiboom–Gil NMR sequence with an overall spin echo time of 150 ms was employed, preceded by a water presaturation pulse during a relaxation delay of 2 s. For each sample, 256 free induction decays of 32 k data points were collected using a spectral width of 6,000 Hz. The free induction decays were multiplied by an exponential weighting function corresponding to a line broadening of 0.3 Hz and zero-filled once before the Fourier transformation. Subsequently, the spectra were phased and baseline-corrected manually and calibrated to the alanine methyl signal ($\delta$ = 1.47 ppm). To facilitate NMR signal assignment, 2D NMR experiments using 1H–1H TOCSY, 1H–13C HSQC, and online databases (Human Metabolome Database, Wishart et al, 2013) were employed. The 1H 1D NMR spectra were directly exported to the AMIX 3.8 software (Bruker Biospin GmbH) and divided into buckets with a width of 0.001 ppm. To remove the effects of variations in the water suppression efficiency, the region between 4.70 and 5.20 ppm was discarded. The obtained NMR dataset X-matrix (40 observations × 8,001 buckets) was then normalized to the total spectral intensity and subjected to multivariate statistical analysis using SIMCA-P + v.16 software (Umetrics). Initially, principal component analysis (PCA) of the 1H NMR spectral data was performed to check the homogeneity of the dataset, group clustering, and detect potential outliers. After PCA, a supervised orthogonal partial least squares discriminant analysis was applied to the X-matrix, in which we defined a Y-matrix as that of sample classes to target metabolic differences between the groups of interest. The resulting score and loading plots were used to visualize the discriminant features. Leave-one-out internal cross-validation was performed to calculate R2Y and Q2 values representing the explained variance of the Y matrix and the predictive ability of the model, respectively. Model validation was performed by random permutation of the Y matrix with n = 999 times and using a CV-ANOVA *P*-value from SIMCA-P + v.16 (analysis of variance in the cross-validated residuals of a Y variable).

**Seahorse**

The OCR rate of in vitro grown MCA205 cells in vitro was measured using a Seahorse XF-24 Metabolic Flux Analyzer. 1 × 10$^5$ cells were

seeded on XF-24 V7 multi-well plates, and then treated or not with 100 $\mu$M Pant for 16 h at 37°C. The OCR was evaluated using an XF Cell Mito Stress Kit (Agilent). $5 \times 10^4$ CD8 TILS isolated from control or Pant-treated murine tumors were analyzed using the T cell Metabolic Profiling kit on Seahorse HS Mini Xfp.i.

### MitoSox

$1 \times 10^5$ MCA205 cells were seeded on 24-well plates in RPMI medium supplemented with 10% FBS and stimulated with 1 mM Pant for 4 h. A 20-min incubation with Antimycin A (5 $\mu$g.ml$^{-1}$) was used as a positive control for mitochondrial ROS production. MitoSox probes were used at 5 $\mu$M for 20 min in pre-warmed HBSS1X. Cells were harvested and MitoSox fluorescence was analyzed by flow cytometry.

### Lactate assay

Lactate concentration was quantified according to the manufacturer's instructions (Sigma-Aldrich). Briefly, the tissues were homogenized in four volumes of lactate assay buffer and centrifuged at 13,000$g$ for 10 min to remove insoluble material. Samples were deproteinized using a 10-kD cut-off spin filter. A master reaction mix containing 46 $\mu$l lactate assay buffer, 2 $\mu$l lactate enzyme mix, and 2 $\mu$l lactate probe was added to 50 $\mu$l sample solution. The reactions were incubated at RT for 30 min and the absorbance of the sample was measured at 570 nm on a microplate reader.

### Flow cytometry and antibodies

For cytometric analysis, tumors were harvested at the indicated time points in DMEM F12 medium, mechanistically dissociated before dissociation using a GENTLEMACS OctoDissociator with heaters from Miltenyi Biotech, according to the manufacturer's protocol. TDLNs were scratched, and mechanically dissociated. After filtration and red blood cell lysis using eBiosciences reagents, the tumor and TDLN cell suspensions were labeled with LiveDead Fixable Blue and CD16/CD32 in PBS EDTA for 30 min. Cell surface labeling was performed in FACS Buffer for 1 h at 4°C with the indicated antibodies. If needed, streptavidin was incubated after cell surface labeling for 15 min at 4°C. For intracellular staining, cells were permeabilized using a FoxP3 staining kit from eBiosciences and antibodies were incubated 45 min at 4°C. Cells were resuspended in FACS Buffer before analysis on BD LSR Symphony or Fortessa. Mitotracker Deep Red and MitroTracker Green were used following the manufacturer's protocol to assess the membrane mitochondrial potential and mitochondrial mass, respectively. The OVA-tetramer (SIINFEKL—H2-K$^b$) was purchased from the NIH tetramer core facility and incubated with cell suspension before cell surface labeling for 1 h at 4°C. Beads were added to the cell suspension for calibration and normalization. Flow cytometry data were analyzed using FlowJo 10.8. The gating strategies for the analysis of myeloid and lymphoid cells are indicated in Figs S1E and S4C, respectively.

### Single-cell RNA sequencing

Tumors from MCA-205-bearing mice treated with PBS or Pant were harvested on D20 or D28 in DMEM F12 medium. After dissociation and RBC lysis, cell suspensions from the four experimental conditions were tagged using the TotalSeq B anti-mouse Hashtag Antibody. CD45-positive live dead negative cells were sorted on ARIA SORP, collected in SVF-coated tubes, and pooled at equivalent ratios. The cell suspensions were immediately processed according to the Chromium Next GEM Single Cell v3.1 GEM protocols. HTO and DNA libraries were sequenced at the CIML genomic facility using a P2 flowcell on an Illumina NextSeq 1000/2000 platform. Two independent experiments (one control versus one Pant representative tumor per experiment) were performed and analyzed separately. FASTQ raw files were aligned to the mouse genome reference (mm10) using 10x Genomics Cell Ranger 6.0.1 (Zheng et al, 2017), which performs filtering, barcode counting, and unique molecular identifier counting. After QC, all count matrices were loaded and processed together using the R package Seurat (6.1.0) (Hao et al, 2021). A second QC was performed to evaluate potential batch-related biases, perform HTO demultiplexing, and identify poor-quality cells to be excluded before downstream analyses. Cells with less than 1,000 counted unique molecular identifiers or more than 5,000 detected genes (potential doublets), and cells with more than 10% mitochondrial gene expression (apoptotic cells) or less than 5% ribosomal genes were excluded from the analyses. Datasets from two independent experiments were pooled and analyzed. Five metaclusters were identified: Myeloid cells, lymphoid cells, APC cells including DC cells and B cells, and an undefined metacluster call "Others" which include tumor cell, fibroblasts, and stromal cells. For subsequent analyses, the metaclusters of the four immune cell types were recomputed separately. After normalization (lognormalize) and centering, the 2,000 most variable features were identified and used for dimensionality reduction by PCA (50 PCs). Further dimensionality reduction was performed using uniform manifold approximation, projection (UMAP) embedding, and cell clustering using the Louvain algorithm (McInnes et al, 2020 Preprint). Marker genes for each cluster were extracted using the Wilcoxon method with the Seurat function "FindMarkers." Functional enrichment analyses were performed using the clusterprofiler (Yu et al, 2012) and fgsea (Korotkevich et al, 2021 Preprint) R packages, and the external database Kyoto Encyclopedia of Genes and Genomes pathways (Kanehisa et al, 2022). GSEA enrichment analyses and figures were performed with the "fgsea" R package against selected reference gmt files from MSigDB (Subramanian et al, 2005; Liberzon et al, 2015). Heatmaps were generated using the R package complex heatmap (Gu et al, 2016) and other custom figures were generated using Wickham (2016). RNA velocity analysis was performed using scVeloAll (Bergen et al, 2020) (detailed parameters for analyses are available in technical reports). For further characterization of NK cells, the results of a differential expression analysis from a previous study (Vienne et al, 2021) comparing ILC and NK cells were downloaded. The top 20 up-regulated and down-regulated genes (sorted by P-value) from this study were used to create gene module scores in Seurat, projected on the previously computed uniform manifold approximation, projection for the NK cell population. RAW sequencing data are available from GEO accession number https://www.ncbi.nlm.nih.gov/geo/query/acc.cgi?acc=GSE221164. The bioinformatics code for analyses and reports generation is hosted on a GitHub repository: https://github.com/CIML-bioinformatic/PNlab_Sarcoma. Resulting data, containers, reports, and figures for analyses of individual samples and merged

datasets are available on Zenodo with DOIs as follows: Miallot et al (2023b, 2023c, 2023d, 2023e, 2023f).

### Cytokine array

Frozen tumors were directly homogenized in PBS-containing protease inhibitors and 1% Triton. After debris removal, protein concentrations were assessed using the BCA assay. Tissue lysates (200 $\mu$g) were processed using the Mouse XL Cytokine Array Kit (R&D Systems). The membranes were scanned with a Samsung Digital Presenter with a 720P HD document camera with a 14x optical zoom and 3x digital zoom image and quantified by Image J.

### CBA

Tumor cell suspensions were centrifuged. TME cytokines were determined in the supernatant using mouse CCL2 and CXCL9 CBA flex set kits (BD Biosciences). Samples were analyzed on a BD LSR II flow cytometer with FCAP Array TM Software V3 (BD Bioscience) to determine cytokine concentrations.

### In vivo depletion

CD8 mAb-based depletion (Clone 53–5.8) was obtained after intraperitoneal injections of 200 and 150 $\mu$g per mouse from day 10 and every 3-d onwards, respectively. A rat IgG1 anti-horseradish mAb (Clone HRPN) was used as an isotype CTRL. NK1.1/ILC depletion was achieved by two injections of anti-NK1.1 mAb (Clone PK136) at 100 $\mu$g per mouse 2 d before MCA205 cell injection and maintained by injections on days 12 and 24. A mouse IgG2a mAb (Clone C1.18.4) was used as the isotype control. The neutralizing IFN$\gamma$ Ultra-LEAF purified antibody was intraperitoneally injected at D20 and D23 at 200 $\mu$g per mouse.

### HSV-1 infection

For HSV-1 infection, the KOS strain of HSV-1 was used (Filtjens et al, 2021). Briefly, HSV-1 was grown and titrated on confluent Vero cells in minimal essential medium containing 10% FCS, 50 $\mu$M 2-mercaptoethanol, 2 mM L-glutamine, 100 U/ml penicillin, and 100 $\mu$g/ml streptomycin. Mice were inoculated with HSV-1 by flank scarification (van Lint et al, 2004). Female mice (8 wk old) were anesthetized by i.p. injection (10 $\mu$l/g body weight) of ketamine (2%)/Rompun (5%) in saline solution. The left flank of each mouse was clipped and depilated with Veet hair remover cream. A small area of the skin was abraded with a MultiPro power tool (Dremel), and 10 $\mu$l of viral suspension containing $10^6$ PFU was applied to the skin lesion. Skin samples were harvested using a 12 × 12 mm skin punch biopsy and immune infiltrate was analyzed on D8 post HSV-1 infection by flow cytometry. To study the virus-specific CD8$^+$ T cell response, we used a tetramer H-2K(b) composed of the HSV-1-derived peptide of the SSIEFARL sequence encoding glycoprotein B (NIH tetramer core facilities).

### DC-OT-1 co-culture

TDLNs from control PBS and Pant mice were collected and CD11c$^+$ cells were sorted using the CD11c Microbeads UltraPure mouse kit (Miltenyi Biotec). CD11c$^+$ cells were incubated in a 96-well plate for 1 h at 37°C with or without the OVA peptide (SIINFEKL). T cells were isolated and purified from the spleens of transgenic OT-1 mice by negative selection using the CD8$\alpha^+$ cell Isolation Kit from Miltenyi Biotec, according to the manufacturer's instructions, and stained with the Cell Trace Violet Cell Proliferation Kit (Invitrogen), according to the manufacturer's instructions. CD11c$^+$ cells and T cells were co-cultured (ratio of 1:10) for 3 d. Cells were stained with a fixable blue dead-cell staining kit (Invitrogen) and anti-mouse CD8 antibody (53-6.7, 55304 BD; PE-Cy5). T cell proliferation was assessed using flow cytometry.

### Immunofluorescence

For immunofluorescence experiments, tumors were harvested, washed, and fixed in Antigenfix solution (Diapath) for 2 h with agitation. After several steps of PBS 1X washing, samples were placed in PBS containing 30% sucrose overnight and included in OCT. After saturation with PBS 2% BSA, samples were stained by overnight incubation at 4°C with anti-CD8b (Clone 53–5.8; BioXcell) and MHC-II (2G9; BD-Pharmingen) mAbs. After incubation with donkey anti-rabbit Cy3 serum and streptavidin Alexa488 (BioLegend), samples were mounted in a mounting medium. Images were acquired using a Zeiss LSM780 confocal microscope and analyzed using ZEN 2.3 software.

### Immunohistology

Histological analysis of osteosarcoma, and MCA tumors was realized on 5-$\mu$m-thick slice mounted in SuperFrost-Plus (VWR). After deparaffinization and rehydration, hematoxylin–phlowin–safran staining was performed to discriminate between the tumor and surrounding tissues. CD8, CD3, and CD31 were stained by immunohistochemistry on the Lyon-Est Research Anatomopathology platform on an automated staining system Ventana Discovery XT (Ventana Medical Systems; Roche). Slides were digitized using Histech 3D slide scanner. The slide scans were analyzed using QuPath v0.3.0 software (Bankhead et al, 2017). After manual annotation of the tumor area, a specific script for IHC and hematoxylin–phlowin–safran staining detection was created to quantify the labeled immune cells and the rate of necrotic tissue present in the tumors. The immune infiltration score displayed in Fig 4B was calculated for each section as follows: 0, cold tumors; 1, peri-infiltrated tumors; 2–3, weak or strong infiltration by TILs in the tumor mass.

### Analysis of *VNN1* expression in STS clinical samples

We analyzed our database (Bertucci et al, 2022) including 1,455 cases of clinically annotated STS clinical samples with available normalized gene expression profiles. *VNN1* expression was available in 1,377 cases that were included in our analyses. Data were gathered from 15 public datasets (Fig S9C), all samples were from operative specimens of previously untreated primary tumors, and the gene expression profiles were generated using DNA microarrays or RNA sequencing. We first searched for robust clusters of co-expressed genes, and the one including *VNN1*, by applying a WGCNA to the 224 TCGA samples, the 8,989 genes filtered based on a SD > 1,

and the following parameters: biweight midcorrelation and a power of 13 to construct a signed gene co-expression network. Ontological analysis of the VNN1-containing cluster was done using the clusterProfiler package, assessing enrichments against the 50 biological functions from the Hallmark database in MSigDB (Yu et al, 2012; Liberzon et al, 2015). Significantly enriched Hallmark functions were defined by an adjusted *P*-value of 5%. Then, *VNN1* mRNA expression was analyzed as discrete variable (high versus low) using two different cut-offs: (i) mean expression level of the whole series: "VNN1-low" if < mean, "VNN1-high" if > mean; (ii) mean expression level of the whole series ±0.5 SD: "VNN1-low-sd" if < mean—0.5 sd, "VNN1-high-sd" if > mean + 0.5 SD. As we found a link between VNN1 and immunity in our mouse model, we searched for correlations between *VNN1* tumor expression and several immunity-related variables. The latter included the following multigene classifiers/ scores: the three Gatza's immune pathway activation scores (Gatza et al, 2010), the 24 Bindea's innate and adaptive immune cell subpopulations (Bindea et al, 2013), two signatures associated with anti-tumor cytotoxic immune response (the Immunologic Constant of Rejection classifier [Bertucci et al, 2022], and Rooney's cytolytic activity score [Rooney et al, 2015]), two metagenes associated with response to ICI (the Ayers' T-cell-inflamed signature (Ayers et al, 2017), and the Coppola's TLS signature, the Ballot's TILs Signature along with both of its inner lymphoid and myeloid signatures (Ballot et al, 2020) and the Thompson's Antigen Processing and presenting Machinery Score (Thompson et al, 2020). The correlations between *VNN1* expression-based classes and clinicopathological variables were measured using Fisher's exact test or *t* test when appropriate. The correlations of immune variables with *VNN1*-based classes were assessed using logistic regression analysis with the glm function (R statistical package; significance estimated by specifying a binomial family for models with a logit link). The endpoint of the prognostic analysis was the MFS, which was calculated from the date of diagnosis to the date of metastatic relapse or death from any cause, whichever occurred first. Follow-up was measured from the date of diagnosis to the date of last news for event-free patients. Survival was estimated using the Kaplan–Meier method, and curves were compared with the log-rank test. Univariate and multivariate prognostic analyses were performed using the Cox regression analysis (Wald test). The variables tested in the univariate analysis were the patients' age and sex, pathological grade (2–3, 1), pathological size (pT1, pT2), tumor depth (deep, superficial), tumor site (extremities, head and neck, internal trunk, superficial trunk), and *VNN1*-based classification (high, low). Multivariate analysis incorporated all variables with a *P*-value < 5% in the univariate analysis. All statistical tests were two-sided, and the significance threshold was set at 5%. Analyses were performed using the survival package (version 2.43) in the R software (version 3.5.2).

### Statistical analysis

The sample sizes were designed to minimize the number of individual experimental units (mice or samples) and obtain informative results and appropriate material for downstream analysis. This represents five mice per group and experiments were typically performed twice, unless otherwise stated in the figure legends.

GraphPad Prism 7 software was used for the statistical significance assessment. The Gaussian distribution was tested using the D'Agostino–Pearson omnibus normality test. When passing the normality test, the *t* test was used. Otherwise, the Mann–Whitney U test was used. Differences were considered statistically significant when ****$P$ < 0.0001, ***$P$ < 0.001, **$P$ < 0.01, *$P$ < 0.05.

# Data Availability

The data used for the bioinformatics analysis were obtained from the publicly available TCGA database. The source code for all the analyses is available at https://github.com/guillaumecharbonnier/ mw-miallot2022 and a compiled version of the Bookdown report is available at https://guillaumecharbonnier.github.io/mw-miallot2022.

# Supplementary Information

# Acknowledgements

The ANR PLBIO19-015 INCA_13754 Grant supported the work realized in the P Naquet, J-C Martin, L Shintu, and J-Y Blay teams and RM funding. We thank the core facilities involved in this project: CIML Flow Cytometry, CIML Genomics Platform, CIML Animal facilities, and Imagimm. B Malissen, and S Henri were supported by the DCBIOL LabEx (grants ANR-11-LABEX-0043 and ANR-10-IDEX-0001-02 PSL). A Roger and S Ugolini were supported by the European Research Council (ERC) under the European Union's Horizon 2020 research and innovation program, under grant agreement No. 648768, from the Agence Nationale de la Recherche (ANR) (No. ANR-14-CE14-00 09-01), and the Fondation pour la Recherche Médicale (FRM). This work was also supported by institutional grants from INSERM, CNRS, Aix-Marseille University, and Marseille-Immunopole to CIML. Guillaume Charbonnier performed bioinformatics analysis of the TCGA database. We thank Thomas Vannier from the CENTURI Multi-Engineering Platform for preliminary work on the data-mining of the cancer databases. The project leading to this publication has received funding from France 2030, the French Government program managed by the French National Research Agency (ANR-16-CONV-0001), and from the Excellence Initiative of Aix-Marseille University—A*MIDEX. J-Y Blay is supported by funds from NetSARC, LYRIC (INCA-DGOS 4664), LYon Recherche Innovation contre le CANcer, European Clinical trials in Rare Sarcomas (FP7-278742), and the European Reference Network for Rare Adult Solid Cancer. We thank A Carrier and T Nguyen for their assistance with the use of the Cell Culture Platform facility (Centre de Recherche contre le Cancer de Marseille, U1068) in charge of the Seahorse platform.

### Author Contributions

R Miallot: data curation, formal analysis, investigation, methodology, and writing—review and editing.
V Millet: resources, data curation, formal analysis, investigation, and methodology.
A Roger: investigation and methodology.
R Fenouil: data curation, software, formal analysis, validation, and methodology.
C Tardivel: methodology.

J-C Martin: resources, formal analysis, validation, investigation, and methodology.

L Shintu: validation, investigation, and methodology.

P Berchard: methodology.

J Sousa Lanza: methodology.

B Malissen: resources.

S Henri: resources.

S Ugolini: resources.

A Dutour: resources.

P Finetti: data curation, software, visualization, and methodology.

F Bertucci: resources, formal analysis, supervision, validation, visualization, and methodology.

J-Y Blay: resources, funding acquisition, validation, and investigation.

F Galland: conceptualization, supervision, visualization, and writing—review and editing.

P Naquet: conceptualization, formal analysis, supervision, funding acquisition, validation, investigation, visualization, project administration, and writing—original draft, review, and editing.

## Conflict of Interest Statement

The authors declare that they have no conflict of interest.

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
