## [Reviewer comments · Life Science Alliance]

Life Science Alliance

The coenzyme A precursor pantethine enhances anti-tumor immunity in sarcoma.

Richard MIALLOT, Virginie Millet, Anais ROGER, Romain Fenouil, Catherine TARDIVEL, Jean Charles Martin, Laetitia Shintu, Paul BERCHARD, Juliane SOUSA LANZA, Bernard Malissen, Sandrine Henri, Sophie Ugolini, Aurelie DUTOUR, Pascal Finetti, François Bertucci, Jean-Yves Blay, Franck Galland, and Philippe Naquet

DOI: <https://doi.org/10.26508/lsa.202302200>

Corresponding author(s): Philippe Naquet, Centre d'Immunologie de Marseille-Luminy and Richard MIALLOT, Centre d'Immunologie de Marseille-Luminy

Review Timeline:

Submission Date:	2023-06-06
Editorial Decision:	2023-07-20
Revision Received:	2023-08-30
Editorial Decision:	2023-09-21
Revision Received:	2023-09-26
Accepted:	2023-09-27

Transaction Report:

July 20, 2023

Re: Life Science Alliance manuscript #LSA-2023-02200-T

Prof. Philippe Naquet
Centre d'Immunologie de Marseille-Luminy
CIML
INSERM-CNRS-Univ. Méditerranée Case 906 Cedex 9
Marseille, PACA 13288
France

Dear Dr. Naquet,

Thank you for submitting your manuscript entitled "The coenzyme A precursor pantethine enhances anti-tumor immunity in sarcoma." to Life Science Alliance. The manuscript was assessed by expert reviewers, whose comments are appended to this letter. We invite you to submit a revised manuscript addressing the Reviewer comments.

Thank you for this interesting contribution to Life Science Alliance. We are looking forward to receiving your revised manuscript.

Sincerely,

B. MANUSCRIPT ORGANIZATION AND FORMATTING:

Reviewer #1 (Comments to the Authors (Required)):

Dear Editor,

The manuscript describes an interesting topic, since especially recently there is building evidence that coenzyme A and coenzyme A precursors play a role in tumour surviving capacity and immunologically related processes. A possible link between vanin expression and tumor survival fits in this novel and still explorative field.

The authors performed numerous experiments using various model systems and analysis, using bioinformatic tools. The manuscript in general does not read well. It is often confusing and the reasoning is not logical. Many models are used, there is overinterpretation of results and it is very descriptive and not focussed.

The reasoning in the introduction towards the research question is not logic to me. The authors mention that VNN1 levels are associated with improved survival of (a limited set) of human sarcoma's. CoA and vitamin B5 have also shown to have some anti-tumor potential.

Vanin hydrolyses pantethine and panteheine into vitamin B5 and cysteamine.

All together one would logically assume that or high vanin levels, or high vitamin B5 levels or high pantethine levels in combination of vanin expression could have anti-tumor properties.

I therefore do not understand the statement made by the authors: " Since VNN1 expression tends to be lost in advanced sarcomas, we speculated that Pant therapy might mimic the beneficial effect of VNN1 expression on anti-tumor immune response in vivo".

It would be more logically to speculate that vitamin B5 in combination with cysteamine might mimic the effect of VNN1 expression. Because VNN1 converts pantethine into vitamin B5 and cysteamine. Or overexpress VNN1 in tumor models in combination with providing pantethine.

The figure legends are also not well explained and are very minimal, as well as the explanation how the bioinformatics were performed.

Because I am not an expert in bioinformatics and statistics, I consulted an expert in this. This person mentioned the following:

Fig 1A

Subdivision in VNN1 high (>5 log₂) vs low (<5 log₂) is too crude: it suggests VNN1 expression (log₂) 5.1 is different from 4.9, which is unlikely. More likely would be a division between VNN1 high (mean + 1 or 2 st.dev) vs VNN1 low (mean - 1 or 2 st.dev)

Fig 1C/D

Why are different patient numbers used for univariate vs multivariate analysis~?

Fig 1B Legend: different N numbers. How many have been used where?

Supplementary S1B and corresponding text:

"we found a significant.....transcripts" (lines 114-116)

Why only focus on the celltypes that do fit the correlation, but disregard the ones that don't?

Suppl. Fig S1C and D and corresponding text:

line 110: "we queried the TCGA database (including 206 soft tissue sarcoma STS samples)....

Line 118: "an independent cohort of 1377 clinical STS samples (figS1C)....

Fig S1C totals 1377 samples, including 265 TGCA samples.

Therefore are these then independent samples? Are they the same, why do the numbers not match (206 vs 265)?

Suppl fig S1D:

Numbers do not match: Female + male= 609, whereas N number age says 532. Follow up/MFS and 5 year MFS indicate N=610

Material and methods (line 692). "we analyzed our database (Bertucci et al., 2022)... Including 1377 samples. That 2022 paper by Bertucci speaks of 1455 samples. What happened to the other samples? Why did they drop out? The usage of which patient where, and for aht reason needs to better described, otherwise it feels like cherry picking to make the data fit the hypothesis.

Fig. 2 C/D

88 C57BL/6 vs 18 nude mice were used. C57BL/6 were not significant until day 18, Nude mice. Similar to nude mice. Difference in nude mice becomes more significant after more days, but nude mice data are not shown after 20 days. The lack of significance in nude, vs presence of significance in BL/6 could be a numbers game. What happens after day 20 in nude mice?

After this the person gave up, it was too much to mention.

Reviewer #2 (Comments to the Authors (Required)):

Miallot et al present a strong manuscript detailing how the administration of the coenzyme A precursor pantethine (a form of Vitamin B5) results in an anti-tumor effect attributed to anti-tumor immunity in a model of sarcoma. This builds on the groups previous work with Vanin1, and appropriate references other papers in the field. The majority of the claims are convincing, with key experiments using immunocompetent vs immunocompromised cancer models and injections of appropriate test compounds. The effect sizes on tumor growth are not huge and a small trend appears in the nude mouse group as well so I think the authors are appropriately circumspect in the abstract. In total, this work is excellent.

Major comments:

1. The second sentence of the abstract is unnecessary and relatively unsupported by later findings. This stems from the assumption that the effects of CoA increase is only mitochondrial is not tested in this manuscript or really well defined in the literature. Specifically, the authors don't examine non-mitochondrial acyl-CoA dependent metabolism so I think they should walk back the claim on the specificity of the mechanism. However, the authors mito experiments in Fig 6 are convincing that the mechanism does include mito-metabolism but to specify that this is only mitochondrial is untested. Specifically, cytoplasmic (Fatty acid synthesis, sterol synthesis), and nuclear (histone acylation) could also be involved.

Minor comments:

1. I thought the pan and cysteamine co injection was a great experiment. Could the authors describe doses in the results, these were very hard to interpret without constantly cross referencing the methods.
2. Spare capacity, not "spared"
3. Fig 2 should add the corresponding number of arrows for the reactions from Pantothenate to CoA. Considering the work on PANKs in cancer metabolism it would be good to give the reader the context that there are additional relevant steps downstream.
4. What are the units on the y-axis of Fig 6A? In the methods, can they specify which ion mode/analysis the panthethine, pantothenate, CoA, and CoA-GSH were detected by metbolon in?
5. CoA-GSH is usually referred to as CoASSG (that's its most common MeSH entry term). The CoA-GSH term is confusing as the GSH is used to refer to the reduced form of glutathione so most literature indicated a disulfide (GSSG, CoASSG) doesn't use that acronym.
6. Can the authors include the whole metabolon dataset rather than picking out a few metabolites without the context of the full reported data set? Just stick it in the supplemental if possible.
7. Can the authors specify the solution the D-pantethine, DCA, cysteamine and pantothenate were resuspended in?

Cross-comments:

The points of reviewers on the TCGA data should be addressed but I understand the majority of the problems there are in the nature of using that data source.

The authors should contact a biostatistician and consider doing a post-hoc power analysis on the nude mouse/immunocompetent mouse comparison, and repeating that experiment as necessary as that is a key experiment fundamental to the claims made in the paper.

I also think the reviewer 1 may have read the originally uploaded version of the manuscript as it was more confusingly written than the updated version.

Dear Editor,

We provide a point-by-point response to referee's requests in blue in the text.

Reviewer #1 (Comments to the Authors (Required)):

The manuscript describes an interesting topic, since especially recently there is building evidence that coenzyme A and coenzyme A precursors play a role in tumour surviving capacity and immunologically related processes. A possible link between vanin expression and tumor survival fits in this novel and still explorative field.

The authors performed numerous experiments using various model systems and analysis, using bioinformatic tools.

The manuscript in general does not read well. It is often confusing and the reasoning is not logical. Many models are used, there is overinterpretation of results and it is very descriptive and not focussed.

The reasoning in the introduction towards the research question is not logic to me. The authors mention that VNN1 levels are associated with improved survival of (a limited set) of human sarcoma's. CoA and vitamin B5 have also shown to have some anti-tumor potential. Vanin hydrolyses pantethine and pantetheine into vitamin B5 and cysteamine.

All together one would logically assume that or high vanin levels, or high vitamin B5 levels or high pantethine levels in combination of vanin expression could have anti-tumor properties.

We agree totally with this statement and reorganized the confusing sentence in the introduction.

I therefore do not understand the statement made by the authors: " Since VNN1 expression tends to be lost in advanced sarcomas, we speculated that Pant therapy might mimic the beneficial effect of VNN1 expression on anti-tumor immune response in vivo".

It would be more logically to speculate that vitamin B5 in combination with cysteamine might mimic the effect of VNN1 expression. Because VNN1 converts pantethine into vitamin B5 and cysteamine. Or overexpress VNN1 in tumor models in combination with providing pantethine.

We have introduced several changes in the text and the legends that hopefully will make the manuscript more understandable (Introduction pages 3-4, result section) and also added explanatory schemes related to the biochemical pathway in Fig2A et supp Fig 11).

The MCA-VNN1 control had been included in FigS4B. We also performed experiments in which pantethine was administered to VNN1 expressing MCA tumors. This strategy did not improve the efficacy of therapy, and generated variability in some experiments that could not be monitored. Indeed, whereas improving CoA levels could be beneficial, it is also known that retroactive metabolic loops can also limit its efficacy and excess pantothenate may also be detrimental and over activate protumor cells as observed in the manuscript when testing mice deprived in cDC1 cells or gIFN pathway (see Fig5A/B).

The figure legends are also not well explained and are very minimal, as well as the explanation how the bioinformatics were performed.

Concerning the bioinformatics analysis, conventional procedures were used and extensively described in the Material and Methods section where all key references are mentioned. Several information have been added (page 19) and provide access to the bioinformatic codes, raw data and results.

Because I am not an expert in bioinformatics and statistics, I consulted an expert in this. This person mentioned the following: **Fig 1A Subdivision in VNN1 high (>5 log2) vs low (<5 log2) is too crude: it suggests VNN1 expression (log2) 5.1 is different from 4.9, which is unlikely. More likely would be a division between VNN1 high (mean + 1 or 2 st.dev) vs VNN1 low (mean - 1 or 2 st.dev).**

VNN1 expression was analyzed as discrete variable (high vs. low) using the mean expression level of the whole series as a cut-off. As suggested by the Reviewer, we tested a second cut-off, more stringent, based on mean \pm 0.5 standard deviation (sd). And the results remain unchanged: we observe the same correlations as the first cut-off between the VNN1 classes and the immune, clinicopathological and MFS data.

This has been added in the revised version as follows:

-In the Materials and Methods section (>): the sentence “VNN1 mRNA expression was analyzed as discrete variable (high vs. low) using the mean expression level as a cut-off.” has been replaced by “VNN1 mRNA expression was analyzed as discrete variable (high vs. low) using two different cut-offs: i) mean expression level of the whole series: “VNN1-low” if $<$ mean, “VNN1-high” if $>$ mean; ii) mean expression level of the whole series \pm 0.5 standard deviation (sd): “VNN1-low-sd” if $<$ mean $-$ 0.5 sd, “VNN1-high-sd” if $>$ mean $+$ 0.5 sd.”.

-In the Results section (page 4): the sentence “VNN1 expression, assessed as discrete variable (high vs. low), correlated (FigS1D) with patient’s age...” has been replaced by the following sentence: “VNN1 expression was assessed as discrete variable (high vs. low) using two different cut-offs. The cut-off based on the mean VNN1 expression value of the whole series allowed analysis of all samples; VNN1 classes correlated (FigS1D) with patient’s age...”. At the end of the first chapter, we have added the results obtained with second cut-off suggested by the reviewer. The following sentences have been added on page 5: “We then applied a second cut-off, more stringent, based on mean \pm 0.5 standard deviation (sd), thus decreasing to 754 the number of patients for correlation analyses (343 “VNN1-high-sd” samples and 411 “VNN1-low-sd” samples), including 344 for MFS analysis. Despite this smaller size, we observed similar correlations between the VNN1 classes and the clinicopathological (FigS2A), immune (FigS2B), and MFS (FigS2C,D) data. The patients with “VNN1-high-sd” samples displayed 63% 5-year MFS (95%CI 55-73) whereas those with “VNN1-low-sd” samples displayed 57% 5-year MFS (95%CI 47-66) (FigS2C). In the univariate prognostic analysis (FigS2D), only VNN1 expression tended towards significance with longer MFS in the patients with “VNN1-high-sd” samples and a HR for event similar to the one observed with the previous cut-off (HR=0.70 95%CI 0.48-1.03, $p=0.072$). » A new supplementary figure (Fig S2) has been added.

-In the Discussion, we have mentioned the use of two different cut-offs giving similar results. On page 13, the sentence « However, when detected, its expression positively correlated with improved prognosis and the presence of immune signatures recapitulating the phenotypes observed in the mouse model » has been replaced by the following sentence : « However, when detected and using two different cut-offs, its expression positively correlated with improved prognosis and the presence of immune signatures recapitulating the phenotypes observed in the mouse model ».

Fig 1C/D: Why are different patient numbers used for univariate vs multivariate analysis?

We thank the Reviewer for his/her comment. There was a small error in the patients included in these analyses. This has been corrected and the results of UV and MV analyses do not change. Our analyses are based on the 610 patients informed in term of MFS. The two tables have been modified as well as the corresponding text in the Results section (pages 5 and 22).

Fig 1B Legend: different N numbers. How many have been used where?

All analyses were done with all 1,377 samples, except for the two modules (Treg* and pDC*) for which the number of genes was not adequate (too low) in 225 samples (1,377 – 1,152). This has been clarified in the figure legend. On page 36, “*N= 1377 or 1152 (*)* » has been replaced by “*N=1,377, except for two modules indicated by “*” (Treg and pDC) that were informed in 1,152 cases*”.

Supplementary S1B and corresponding text: "we found a significant.....transcripts" (lines 114-116); Why only focus on the cell types that do fit the correlation, but disregard the ones that don't?
Suppl. Fig S1C and D and corresponding text: line 110: "we queried the TCGA database (including 206 soft tissue sarcoma STS samples).... Line 118: "an independent cohort of 1377 clinical STS samples (figS1C)....

Fig S1C totals 1377 samples, including 265 TCGA samples. Therefore are these then independent samples? Are they the same, why do the numbers not match (206 vs 265)?

We agree with the Reviewer that there was a redundancy (TCGA sets, immune Cibersort and Bindea modules) and inconsistency in the number of TCGA samples in analyses shown in Figures S1A-B and Figure 1B. Thus, we have deleted the previous Weighted Gene Correlation Network Analysis (WGCNA) done in the 206 TCGA samples, and we redid it in the 224 TCGA STS samples included in our 1,377 cases. The new WGCNA analysis (224 samples) gives the same results as the previous one. We have also deleted the previous CIBERSORT analysis done on 206 TCGA STS samples (redundancy with Bindea's modules analysis shown in Figure 1B).

The following modifications have been done:

-deletion of the “Analysis of VNN1 expression in TCGA Database and Cibersort” chapter on page 21 in the Material & Methods section.

-addition of the following sentences in the “Analysis of VNN1 expression in soft tissue sarcoma clinical samples” chapter on page 21 in the Material & Methods section: “*We first searched for robust clusters of co-expressed genes, and the one including VNN1, by applying a Weighted Gene Correlation Network Analysis (WGCNA) to the 224 TCGA samples, the 8,989 genes filtered based on a standard deviation > 1, and the following parameters: biweight midcorrelation and a power of 13 to construct a signed gene co-expression network. Ontological analysis of the VNN1-containing cluster was done using the clusterProfiler package, assessing enrichments against the 50 biological functions from the Hallmark database in MSigDB (Yu et al., 2012; Liberzon et al., 2015). Significantly enriched Hallmark functions were defined by an adjusted p-value of 5%*”.

-in the Results section, the following sentences in the “Analysis of VNN1 expression in soft tissue sarcoma clinical samples” chapter on page 4: “*We queried the TCGA database (including 206 soft tissue sarcoma STS samples) to search for VNN1-coexpressed gene modules. VNN1 expression matched MSigDB hallmarks associated with IFN γ /inflammatory responses (FigS1A). Using a CIBERSORT (<http://cibersort.stanford.edu/>) deconvolution method to link VNN1 expression with the presence of immunocyte subsets, we found a significant correlation between high VNN1 expression and the presence of immune cells that do not express VNN1 such as M1/M2 macrophages and T cells (including CD8+) transcripts (FigS1B).*” have been replaced by the following sentences: “*We first queried the 224 TCGA STS samples using the Weighted Gene Correlation Network Analysis (WGCNA) to define robust gene clusters and to search for the cluster of genes co-expressed with VNN1. WGCNA revealed 8 gene clusters including a 505-gene cluster containing VNN1 (FigS1A), Ontology analysis showed that this cluster was strongly associated with immune MSigDB hallmarks, such as allograft rejection, inflammatory responses or IFN γ response (FigS1B).* ». The figure S1A-B has been modified accordingly.

Suppl fig S1D: Numbers do not match: Female + male= 609, whereas N number age says 532. Follow up/MFS and 5 year MFS indicate N=610.

The missing data represent a known bias of retrospective studies. The following sentence has been added in the Discussion on page 13: “*Such in silico analyses display strengths and limitations. The first*

limitation is related to the retrospective nature and its associated biases such as heterogeneity and missing data. For example, survival data were available for 610 STS patients and age for 532 patients. However to our knowledge, our series is the largest one analyzing VNN1 expression in STS. Other strengths include its originality, being the first one to describe the clinicopathological and molecular correlations with VNN1 expression in STS samples, and the use of a patients' population treated in the community setting, likely more reflective of real-life than a selected patient population within a clinical trial."

Material and methods (line 692). "we analyzed our database (Bertucci et al., 2022)... Including 1377 samples. That 2022 paper by Bertucci speaks of 1455 samples. What happened to the other samples? Why did they drop out? The usage of which patient where, and for that reason needs to be better described, otherwise it feels like cherry picking to make the data fit the hypothesis.

In fact, VNN1 expression was available in 1,377 out of 1,455 cases that were included in our analyses. This has been clarified on page 21. The sentence "*We analyzed our database (Bertucci et al., 2022) including 1.377 cases of clinically annotated STS clinical samples with available normalized gene expression profiles.*" has been replaced by the following sentence: "*We analyzed our database (Bertucci et al., 2022) including 1.455 cases of clinically annotated STS clinical samples with available normalized gene expression profiles. VNN1 expression was available in 1,377 cases that were included in our analyses* ».

Fig. 2 C/D. 88 C57BL/6 vs 18 nude mice were used. C57BL/6 were not significant until day 18, Nude mice. Similar to nude mice. Difference in nude mice becomes more significant after more days, but nude mice data are not shown after 20 days. The lack of significance in nude, vs presence of significance in BL/6 could be a numbers game. What happens after day 20 in nude mice?

We have introduced several modifications in figure 2D and its legend (page 36): first an additional experiment was pooled to the two first ones bringing the number of samples from 18 to 26 for the nude mouse experiment. Second, we added the size of the tumors at later time points (up to day 22 in nude animals) to match the data presented in wild type mice. The reason why we could not include the complete set of data beyond day 22 is linked to the fact that the size of tumors growing in nude mice beyond day 18 often exceeds the limit authorized by legislation. As it can be seen in the reformatted figure, there is no difference with and without pantothenate injection in the nude mouse model.

Reviewer #2 (Comments to the Authors (Required)):

Miallot et al present a strong manuscript detailing how the administration of the coenzyme A precursor pantothenine (a form of Vitamin B5) results in an anti-tumor effect attributed to anti-tumor immunity in a model of sarcoma. This builds on the groups previous work with Vanin1, and appropriate references other papers in the field. The majority of the claims are convincing, with key experiments using immunocompetent vs immunocompromised cancer models and injections of appropriate test compounds. The effect sizes on tumor growth are not huge and a small trend appears in the nude mouse group as well so I think the authors are appropriately circumspect in the abstract. In total, this work is excellent.

Major comments:

1. The second sentence of the abstract is unnecessary and relatively unsupported by later findings.

We omitted this sentence (page 2) but also completed the information extracted from the metabolomics analysis that shows variations in other lipid metabolites (page 9, FigS12 and comments below)

This stems from the assumption that the effects of CoA increase is only mitochondrial is not tested in this manuscript or really well defined in the literature. Specifically, the authors don't examine non-mitochondrial acyl-CoA dependent metabolism so I think they should walk back the claim on the specificity of the mechanism. However, the authors mito experiments in Fig 6 are convincing that the mechanism does include mito-metabolism but to specify that this is only mitochondrial is untested. Specifically, cytoplasmic (Fatty acid synthesis, sterol synthesis), and nuclear (histone acylation) could also be involved.

We agree that some aspects of metabolic pathways have not been investigated in depth. Nevertheless, we have included all the values presented as significant by Metabolon analysis in an additional FigS12 and discussed on page 9. This includes data concerning several lipid metabolites associated with beta oxidation (in general reduced in the presence of pantethine, in agreement with increased consumption during lipid catabolism) or phospholipid synthesis (in general augmented by pantethine and in agreement with the idea that cell fitness is globally enhanced by this drug). We have not explored histone acetylation marks but the transcriptomic and qRT-PCR analysis showed that variations in gene expression were of modest amplitude and concerned limited genesets.

Minor comments:

1. I thought the pan and cysteamine co injection was a great experiment. Could the authors describe doses in the results, these were very hard to interpret without constantly cross referencing the methods.

The doses were indicated in the material and methods section (see on page 14). Pantethine can be injected at high doses without toxicity (1g/kg which corresponds to 45 micromoles per injection per mouse). Although pantethine effect could already be observed at 500 mg/kg (not shown), we had opted for this higher dose to optimize the pharmacological effect of the compound as shown in several publications cited in the text. Such doses cannot be obtained for cysteamine which is toxic beyond 120 mg/kg (6 micromoles per injection per mouse). The pharmacological effect of the coinjection of cysteamine and pantothenate mirrored that of pantethine in our experiments.

2. Spare capacity, not "spared"

Corrected

3. Fig 2 should add the corresponding number of arrows for the reactions from Pantothenate to CoA. Considering the work on PANKs in cancer metabolism it would be good to give the reader the context that there are additional relevant steps downstream.

We completed the information provided in Fig2A and added more complete information in FigS11. We also added 2 references discussing the links between PANK levels and cancer risk in the discussion. See on page 13.

4. What are the units on the y-axis of Fig 6A? In the methods, can they specify which ion mode/analysis the panthethine, pantothenate, CoA, and CoA-GSH were detected by metbolon in?

To remove batch variability, for each metabolite, the values in the experimental samples are divided by the median of those samples in each instrument batch, giving each batch and thus the metabolite a median of one. For each metabolite, the minimum value across all batches in the median scaled data is imputed for the missing values. This information has been added in the Material and Methods section (page 15).

The Batch-norm-Imputed Data is transformed using the natural log. Metabolomic data typically displays a log-normal distribution, therefore, the log-transformed data is used for statistical analyses.

5. CoA-GSH is usually referred to as CoASSG (that's its most common MeSH entry term). The CoA-GSH term is confusing as the GSH is used to refer to the reduced form of glutathione so most literature indicated a disulfide (GSSG, CoASSG) doesn't use that acronym.

Corrected

6. Can the authors include the whole metabolon dataset rather than picking out a few metabolites without the context of the full reported data set? Just stick it in the supplemental if possible.

We have included and discussed (see page 9) all the values presented as significant by Metabolon analysis in an additional FigS12. Full data is accessible in data source provided as an excel file.

7. Can the authors specify the solution the D-pantethine, DCA, cysteamine and pantothenate were resuspended in?

Concentrated stock solutions were prepared in water and diluted in PBS for injections.

Cross-comments:

The points of reviewers on the TCGA data should be addressed but I understand the majority of the problems there are in the nature of using that data source.

The authors should contact a biostatistician and consider doing a post-hoc power analysis on the nude mouse/immunocompetent mouse comparison, and repeating that experiment as necessary as that is a key experiment fundamental to the claims made in the paper.

We have introduced several modifications in this figure (see corrected Fig2D): first an additional experiment was pooled to the two first ones bringing the number of samples from 18 to 26, providing enough statistical power. Second, we added the size of the tumors at later time points to match the data presented in wild type mice and enhance the strength of the results. The reason why we had not included the complete set of data in this first submission was linked to the fact that the size of tumors growing in nude mice beyond day 18 often exceeds the limit authorized by legislation. As it can be seen in the reformatted figure, there is no difference with and without pantetheine injection in the nude mouse model.

I also think the reviewer 1 may have read the originally uploaded version of the manuscript as it was more confusingly written than the updated version.

September 21, 2023

RE: Life Science Alliance Manuscript #LSA-2023-02200-TR

Prof. Philippe NAQUET
Centre d'Immunologie de Marseille-Luminy
CIML
INSERM-CNRS-Univ. Méditerranée Case 906 Cedex 9
Marseille, PACA 13288
France

Dear Dr. Naquet,

Thank you for submitting your revised manuscript entitled "The coenzyme A precursor pantethine enhances anti-tumor immunity in sarcoma.". We would be happy to publish your paper in Life Science Alliance pending final revisions necessary to meet our formatting guidelines.

- please upload your main manuscript text as an editable doc file
- please add ORCID ID for secondary corresponding author--they should have received instructions on how to do so
- please add a Category for your manuscript in our system
- please add the Twitter handle of your host institute/organization as well as your own or/and one of the authors in our system
- please rename your section "Availability of data and materials" to "Data Availability"
- please use the [10 author names, et al.] format in your references (i.e. limit the author names to the first 10)
- please add a callout for Figure S7B to your main manuscript text
- please note that legends should appear only in manuscript text, remove them from the figures
- please exclude figures from the manuscript text

Figure checks:

- please add scale bars to Figure S4C and S6E

A. FINAL FILES:

B. MANUSCRIPT ORGANIZATION AND FORMATTING:

Sincerely,

Reviewer #2 (Comments to the Authors (Required)):

The authors adequately responded to my comments. I look forward to seeing the work published.

September 27, 2023

RE: Life Science Alliance Manuscript #LSA-2023-02200-TRR

Prof. Philippe Naquet
Centre d'Immunologie de Marseille-Luminy
CIML
INSERM-CNRS-Univ. Méditerranée Case 906 Cedex 9
Marseille, PACA 13288
France

Dear Dr. Naquet,

Thank you for submitting your Research Article entitled "The coenzyme A precursor pantethine enhances anti-tumor immunity in sarcoma.". It is a pleasure to let you know that your manuscript is now accepted for publication in Life Science Alliance. Congratulations on this interesting work.

DISTRIBUTION OF MATERIALS:

Again, congratulations on a very nice paper. I hope you found the review process to be constructive and are pleased with how the manuscript was handled editorially. We look forward to future exciting submissions from your lab.

Sincerely,
